# Better Linear Rates for SGD with Data Shuffling

## Abstract

Virtually all state-of-the-art methods for training supervised machine learning models are variants of SGD, enhanced with a number of additional tricks, such as minibatching, momentum, and adaptive stepsizes. However, one of the most basic questions in the design of successful SGD methods, one that is orthogonal to the aforementioned tricks, is the choice of the *next* training data point to be learning from. Standard variants of SGD employ a *sampling with replacement* strategy, which means that the next training data point is sampled from the entire data set, often independently of all previous samples. While standard SGD is well understood theoretically, virtually all widely used machine learning software is based on *sampling without replacement* as this is often empirically superior. That is, the training data is randomly shuffled/permuted, either only once at the beginning, strategy known as *random shuffling* (Rand-Shuffle), or before every epoch, strategy known as *random reshuffling* (Rand-Reshuffle), and training proceeds in the data order dictated by the shuffling. RS and RR strategies have for a long time remained beyond the reach of theoretical analysis that would satisfactorily explain their success. However, very recently, Mishchenko et al. (2020) provided tight *sublinear* convergence rates through a novel analysis, and showed that these strategies can improve upon standard SGD in certain regimes. Inspired by these results, we seek to further improve the rates of shuffling-based methods. In particular, we show that it is possible to enhance them with a variance reduction mechanism, obtaining *linear* convergence rates. To the best of our knowledge, our linear convergence rates are the best for any method based on sampling without replacement.

## 1 Introduction

The main paradigm for training supervised machine learning models—Empirical Risk Minimization (ERM)—is an optimization problem of the finite sum structure

$$\min_{x \in \mathbb{R}^d} \left[ f(x) := \frac{1}{n} \sum_{i=1}^{n} f_i(x) \right], \tag{1}$$

where $x \in \mathbb{R}^d$ is a vector representing the parameters (model weights, features) of a model we wish to train, $n$ is the number of training data points, and $f_i(x)$ represents the (smooth) loss of the model $x$ on data point $i$. The goal of ERM is to train a model whose average loss on the training data is minimized. This abstraction allows to encode virtually all supervised models trained in practice, including linear and logistic regression, and neural networks.

The gigantic size of modern training data sets necessary to train models with good generalization poses severe issues for the designers of methods for solving equation 1. Over the last decade, stochastic first-order methods have emerged as the methods of choice, and for this reason, their importance in machine learning remains exceptionally high (Bottou et al., 2018). Of these, stochastic gradient descent (SGD) is perhaps the best known, but also the most basic. SGD has a long history (Robbins & Monro, 1951; Bertsekas & Tsitsiklis, 1996) and is therefore well-studied and well-understood (Rakhlin et al., 2012; Hardt et al., 2016; Drori & Shamir, 2019; Gower et al., 2019; Nguyen et al., 2020).

**Training data order.** Standard and even variance-reduced variants of SGD employ a *sampling with replacement* strategy (Gorbunov et al., 2020), which means that the next training data point in each epoch is sampled from the entire data set, independently of all previous samples. However, virtually all widely used machine learning software is based on *sampling without replacement* as this is often empirically superior (Bottou, 2009; Recht & Ré, 2013), and therefore

acts as the de-facto default sampling mechanism in deep learning (Bengio, 2012; Sun, 2020). With this latter strategy, in each epoch we sample each training data exactly once, and this can be performed by generating a random permutation of the training data.

There are three commonly used variants of sampling without replacement.

(i) In the first, which we call *deterministic shuffling* (Det-Shuffle) in this paper, the training data is processed in some natural order in a cyclic manner. That is, a deterministic permutation is used throughout the entire training process. This idea is the basis of the Cyclic-GD method (Luo, 1991; Grippo, 1994). While this strategy is not effective in practice, it is perhaps the simplest strategy conceptually, and has been studied repeatedly. However, it is notoriously difficult to obtain good guarantees for it.

(ii) In the second variant, which we call *random shuffling* (Rand-Shuffle) in this paper[1], the training data is instead shuffled/permuted randomly. This is done only once, before the start of the training process, and the selection of training data then follows a cyclic pattern dictated by this single random permutation (Nedić & Bertsekas, 2001). The purpose of this procedure is to break the potentially adversarial default ordering of the data that could negatively affect training speed. Almost no non-trivial analyses exist for this method (Mishchenko et al., 2020). This strategy works very well in practice.

(iii) In the third variant, known as *random reshuffling* (Rand-Reshuffle), the training data is randomly reshuffled before the start of each epoch. This is perhaps the most common and relatively most studied approach. Its empirical performance is, however, often very similar to Rand-Shuffle, and the current best theoretical bounds for both are the same (Mishchenko et al., 2020).

**Difficulties with analyzing shuffling-based methods.** The main difficulty in analyzing methods based on sampling without replacement is that each gradient step within an epoch is *biased*, and performing a sharp analysis of methods based on biased estimators is notoriously difficult. While Cyclic-GD was studied already a few decades ago (Mangasarian & Solodov, 1994; Bertsekas & Tsitsiklis, 2000), convergence rates were established relatively recently (Li et al., 2019; Ying et al., 2019; Gürbüzbalaban et al., 2019; Nguyen et al., 2020). For the Rand-Shuffle method, the situation is more complicated, and non-vacuous theoretical analyses were only performed recently (Safran & Shamir, 2020; Rajput et al., 2020). Rand-Reshuffle is well understood for twice-smooth (Gürbüzbalaban et al., 2019; Haochen & Sra, 2019) and smooth (Nagaraj et al., 2019) objectives. Moreover, lower bounds for Rand-Reshuffle and similar methods were also recently established (Safran & Shamir, 2020; Rajput et al., 2020). Mishchenko et al. (2020) recently performed an in-depth analysis of Det-Shuffle, Rand-Shuffle and Rand-Reshuffle with novel and simpler proof techniques, leading to improved and new convergence rates. Their rate for Rand-Shuffle, for example, tightly matches the lower bound of Safran & Shamir (2020) in the case when each $f_i$ is strongly convex. Further, Rand-Reshuffle can be accelerated (Gürbüzbalaban et al., 2019), and for small constant step-sizes, the neighborhood of solution can be controlled (Sayed, 2014). However, despite these advances, Rand-Reshuffle and related method described above still suffer from the same problem as SGD, i.e., we do not have variants that would have a fast linear convergence rate to the exact minimizer.

**Variance reduction.** Despite its simplicity and elegance, SGD has a significant disadvantage: the variance of naive stochastic gradient estimators of the true gradient remains high throughout the training process, which causes issues with convergence. When a constant learning rate is used in the smooth and strongly convex regime, SGD converges linearly to a neighborhood of the optimal solution of size proportional to the learning rate and to the variance of the stochastic gradients at the optimum (Gower et al., 2020). While a small or a decaying learning schedule restores convergence, the convergence speed suffers as a result. Fortunately, there is a remedy for this ailment: *variance-reduction* (VR) (Johnson & Zhang, 2013). The purpose of VR mechanisms is to steer away from the naive gradient estimators. Instead, VR mechanisms iteratively construct and apply a gradient estimator whose variance would eventually vanish. This allows for larger learning rates to be used safely, which accelerates training. Among the early VR-empowered SGD methods belong SAG (Roux et al., 2012), SVRG (Johnson & Zhang, 2013), SAGA (Defazio et al., 2014a), and Finito (Defazio et al., 2014b). For a recent survey of VR methods, see (Gower et al., 2020).

**Related work.** Some cyclic and random reshuffling versions of variance-reduced methods were shown to obtain linear convergence. Incremental Average Gradient (IAG)—a cyclic version of the famous SAG method—was analyzed by Gürbüzbalaban et al. (2017). Based on this, the Doubly Incremental Average Gradient (DIAG) method was introduced,

---

[1]This method is called "shuffle once" in some papers.

and it has a significantly better rate if each $f_i$ is strongly convex (Mokhtari et al., 2018). A linear rate for Cyclic-SAGA was established by Park & Ryu (2020). The first analysis of Rand-Reshuffle with variance reduction was done by Ying et al. (2020). Firstly, they establish a linear rate for SAGA under random reshuffling, and then they introduce a new method called Amortized Variance-Reduced Gradient (AVRG), which is similar to SAGA. SVRG using Rand-Reshuffle was introduced by Shamir (2016), and their theoretical analysis was conducted for the Least Squares problem. The promising result of Prox-DFinito is introduced in Huang et al. (2021) for the composite optimization problem.

## 2 Approach and Contributions

Let us now briefly outline our approach and key contributions.

### 2.1 Controlled linear perturbations

In the design of our methods we employ a simple but powerful tool: the idea of introducing a sequence of carefully crafted reformulations of the original finite sum problem, and applying vanilla shuffling-based methods on these reformulations instead of the original formulation. As the sequence is designed to have progressively better conditioning properties, our methods will behave progressively better as well, and this is why this result in variance reduced shuffling methods.

The main idea is to perturb the objective function with zero written as the average of $n$ nonzero linear functions. This perturbation is performed at the beginning of each epoch, and stays fixed within each epoch. Let us consider the finite sum problem equation 1 and vectors $a_t^i, \ldots, a_t^n \in \mathbb{R}^d$ summing up to zero: $\sum_{i=1}^n a_t^i = 0$. Let $a^t = (a_t^i, \ldots, a_t^n)$. Adding this *structured* zero to $f$, we reformulate problem equation 1 into the equivalent form

$$f(x) = \frac{1}{n} \sum_{i=1}^n f_i(x) = \frac{1}{n} \sum_{i=1}^n \left( f_i(x) + \left\langle a_t^i, x \right\rangle \right) := \frac{1}{n} \sum_{i=1}^n f_i^t(x), \tag{2}$$

where $f_i^t(x) := f_i(x) + \left\langle a_t^i, x \right\rangle$. Note that

$$\nabla f_i^t(x) = \nabla f_i(x) + a_t^i. \tag{3}$$

Next, we establish a simple but important property of this reformulation.

**Proposition 1.** *Assume that each $f_i$ is $\mu$-strongly convex (resp. convex) and $L$-smooth. Then $f_i^t$ is $\mu$-strongly convex (resp. convex) and $L$-smooth.*

In our methods, the vectors $a_t^1, \ldots, a_t^n$ depend on two objects:

- a *control vector* $y_t \in \mathbb{R}^d$, which is updated at the start of each epoch,

- the *permutation* $\pi = \{\pi_0, \pi_1, \ldots, \pi_{n-1}\}$ chosen at the beginning of the current epoch.

In particular, we choose

$$a_t^i := -\nabla f_{\pi_i}(y_t) + \nabla f(y_t). \tag{4}$$

Note that by plugging equation 4 into equation 3, the gradient of $f_{\pi_i}^t$ at $x \in \mathbb{R}^d$ is given by

$$g_t^i(x, y_t) := \nabla f_{\pi_i}(x) - \nabla f_{\pi_i}(y_t) + \nabla f(y_t). \tag{5}$$

At the start of each epoch, the control vector $y_t$ is set to the latest iterate $x_t$.

### 2.2 New algorithms: improvement of shuffling based methods

Our key proposal is to run *standard* Det-Shuffle, Rand-Shuffle and Rand-Reshuffle methods, for example as described in (Mishchenko et al., 2020), but in each epoch to apply them to the current reformulated problem

$$\min_{x \in \mathbb{R}^d} \frac{1}{n} \sum_{i=1}^n f_i^t(x).$$

---

**Algorithm 1** Algorithms Det-Shuffle, Rand-Shuffle, Rand-Reshuffle

---

**Input:** Stepsize $\gamma > 0$, initial iterate $x_0 \in \mathbb{R}^d$, number of epochs $T$

**Option** Det-Shuffle: Choose a deterministic permutation $\{\pi_0, \ldots, \pi_{n-1}\}$ of $\{1, \ldots, n\}$

**Option** Rand-Shuffle: Choose a random permutation $\{\pi_0, \ldots, \pi_{n-1}\}$ of $\{1, \ldots, n\}$

**for** $t = 0, 1, \ldots T - 1$ **do**

    **Option** Rand-Reshuffle: Choose a random permutation $\{\pi_0, \ldots, \pi_{n-1}\}$ of $\{1, \ldots, n\}$

    $x_t^0 = x_t, y_t = x_t$

    **for** $i = 0, \ldots, n - 1$ **do**

        $g_t^i(x_t^i, y_t) = \nabla f_{\pi_i}(x_t^i) - \nabla f_{\pi_i}(y_t) + \nabla f(y_t)$

        $x_t^{i+1} = x_t^i - \gamma g_t^i(x_t^i, y_t)$

    **end for**

    $x_{t+1} = x_t^n$

**end for**

---

This leads to our variance-reduced algorithms, all described compactly in Algorithm 1. Hoping that this will not cause confusion, we do not give the methods a different name.

- Note that as mentioned in the introduction, in Det-Shuffle we only use a single deterministic permutation at the start of the method. The steps are then performed incrementally through all data, in the same order in each epoch.

- In contrast, in Rand-Shuffle we shuffle the data points randomly instead, but otherwise proceed as in Det-Shuffle, using this one permutation in all subsequent epochs.

- Finally, Rand-Reshuffle is similar to Rand-Shuffle, with the exception that a new permutation is resampled at the start of each epoch.[2]

Besides Algorithm 1, we also propose a generalized version of Rand-Reshuffle (Algorithm 2), which differs from Rand-Reshuffle in that at the end of each epoch we flip a biased coin to decide whether to update the control vector $y_t$ or not. While in Rand-Reshuffle the control vector $y_{t+1}$ is updated to the latest iterate $x_{t+1}$, in Algorithm 2 we use the previous point $x_t$. We do this as it slightly simplified the analysis. However, it makes sense to use the newest point $x_{t+1}$ instead of $x_t$ to update the control vector in practice. This method is described in the appendix only.

### 2.3 Analysis technique: the basic idea

Since in view of Proposition 1 the reformulated problem satisfies all assumptions of the original problem, in a single epoch it is possible to apply results that hold for vanilla Det-Shuffle, Rand-Shuffle and Rand-Reshuffle methods – variants that are not variance-reduced. In particular, we rely on some results of Mishchenko et al. (2020), and complement them with new analysis that handles the changing nature of the reformulations through the change in the control vectors $\{y_t\}$.

In particular, a key insight of our paper is the observation that by updating the control vector, we can control the variance of shuffling based methods.[3]

We are now ready to formulate the core lemma of our work.

**Lemma 1.** *Assume that each $f_i$ is $L$-smooth and convex. If we apply the linear perturbation reformulation equation 2 using vectors of the form equation 4, then the gradient variance of the reformulated problem at the optimum $x_*$ can be bounded via the distance of the control vector $y_t$ to $x_*$ as follows:*

$$(\sigma_*^t)^2 := \frac{1}{n} \sum_{i=1}^{n} \left\| \nabla f_i^t(x_*) \right\|^2 \leq 4L^2 \|y_t - x_*\|^2. \tag{6}$$

Table 1: Complexity of shuffling based methods (in all expressions we ignore constant terms).

| Algorithm | $\mu$-strongly convex $f_i$ | $\mu$-strongly convex $f$ | convex $f$ | non-convex $f$ | memory | reference |
|---|---|---|---|---|---|---|
| RR-SAGA | – | $\kappa^2 \log 1/\epsilon$ | – | – | $dn$ | Ying et al. (2020) |
| AVRG | – | $\kappa^2 \log 1/\epsilon$ | – | – | $d$ | Ying et al. (2020) |
| Rand-Shuffle Rand-Reshuffle | $\kappa \sqrt{\tfrac{\kappa}{n}} \log 1/\epsilon$ [1] | $\kappa \log 1/\epsilon$ [1] $\kappa \sqrt{\kappa} \log 1/\epsilon$ [2] | $L/\epsilon$ | $L/\epsilon^2$ [5] | $d$ | this paper |
| Prox-DFinito | $\kappa \log 1/\epsilon$ | – | $L^2/\epsilon$ | – | $dn$ | Huang et al. (2021) |
| Cyclic-SAGA | $\kappa^2 \log 1/\epsilon$ | – | – | – | $dn$ | Park & Ryu (2020) |
| IAG [3] | – | $n\kappa^2 \log 1/\epsilon$ | – | – | $dn$ | Gürbüzbalaban et al. (2017) |
| DIAG [4] | $\kappa \log 1/\epsilon$ | – | – | – | $dn$ | Mokhtari et al. (2018) |
| Det-Shuffle | – | $\kappa \sqrt{\kappa} \log 1/\epsilon$ | $L/\epsilon$ | – | $d$ | this paper |

[1] Big data regime.
[2] General regime.
[4] Cyclic version of the Stochastic Average Gradient (SAG) method, which was the original inspiration for SAG.
[3] Cyclic version of the Finito algorithm.
[5] The result is applied to Rand-Reshuffle

## 2.4 Complexity results

Our theory leads to improved rates for shuffling-based methods using all three sampling strategies: Det-Shuffle, Rand-Shuffle and Rand-Reshuffle. We provide theoretical guaranties in Section 3; a summary is presented in Table 1.

◇ **Strongly convex case.** If $f$ is strongly convex, we obtain $O\left(\kappa^{3/2} \log 1/\varepsilon\right)$ iteration (epoch-by-epoch) complexity for Rand-Reshuffle, where $\kappa$ is the condition number. This rate is better than the $O\left(\kappa^2 \log 1/\varepsilon\right)$ rate of RR-SAGA and AVRG introduced by Ying et al. (2020). Moreover, if $n > O(\kappa)$, we improve this rate for Rand-Reshuffle and get $O\left(\kappa \log 1/\varepsilon\right)$ complexity. If each $f_i$ is strongly convex and the number of functions is sufficiently large (Theorem 3), then the rate of Rand-Reshuffle can be further improved to $O(\kappa \sqrt{\kappa/n} \log 1/\varepsilon)$. For Det-Shuffle we prove similar convergence results under the assumption of strong convexity of $f$. The iteration complexity of this method is $O\left(\kappa^{3/2} \log 1/\varepsilon\right)$, which is noticeably better than the $O\left(n\kappa^2 \log 1/\varepsilon\right)$ rate of IAG (Gürbüzbalaban et al., 2017). Furthermore, it is better than the $O\left(\kappa^2 \log 1/\varepsilon\right)$ rate of Cyclic-SAGA (Park & Ryu, 2020). It is worth mentioning that Mokhtari et al. (2018) obtain a better complexity, $O\left(\kappa \log 1/\varepsilon\right)$, for their DIAG method. However, their analysis requires much stricter assumption.

◇ **Convex case.** In the general convex setting we give the first analysis and convergence guarantees for Det-Shuffle, Rand-Shuffle, and Rand-Reshuffle. After applying variance reduction, we obtain fast convergence to the exact solution. As expected, these methods have the sublinear rate $O(\frac{1}{\varepsilon})$ in an ergodic sense.

## 2.5 Shuffling-based variants of variance reduced methods.

While, as we argue, our methods should be seen as improvements over existing shuffling-based methods via variance reduction, it is possible to alternatively see them as shuffling-based variants of variance reduced methods. However, when seen that way, we do not observe an improvement in complexity. The reason for this is that there is a large gap in our understanding of shuffling based methods, especially for variance reduced variants, which does not yet allow for theoretical speedups compared to their sampling-with-replacement cousins. For example, from the latter viewpoint, and to the best of our knowledge, we provide the first convergence analysis of SVRG under random reshuffling. However, the rate of classical variance reduced methods, such as SVRG, is still superior in some regimes.

# 3 Main Theoretical Results

Having described the methods and the idea of controlled linear perturbations, we are ready to proceed to the formal statement of our convergence results.

---

[2] Note that Rand-Reshuffle can be seen as a version of SVRG in which the number of inner steps $m$ is equal to $n$, and in which sampling *without* replacement is used. Johnson & Zhang (2013) remarked that $m = O(n)$ works well in practice, but a theoretical analysis of this was not provided.

[3] While this was known for methods based on sampling with replacement, this is a new observation for methods based on sampling without replacement, and our control strategy.

### 3.1 Assumptions and Notation

Before introducing our convergence results, let us first formulate the definitions and assumptions we use throughout the work. Function $f : \mathbb{R}^d \to \mathbb{R}$ is $L$-smooth if

$$f(y) \le f(x) + \langle \nabla f(x), y - x \rangle + \frac{L}{2} \|y - x\|^2 \quad \forall x, y \in \mathbb{R}^d, \tag{7}$$

convex if

$$f(x) + \langle \nabla f(x), y - x \rangle \le f(y) \quad \forall x, y \in \mathbb{R}^d, \tag{8}$$

and $\mu$-strongly convex if

$$f(x) + \langle \nabla f(x), y - x \rangle + \frac{\mu}{2} \|y - x\|^2 \le f(y) \quad \forall x, y \in \mathbb{R}^d. \tag{9}$$

The Bregman divergence with respect to $f$ is the mapping $D_f : \mathbb{R}^d \times \mathbb{R}^d \to \mathbb{R}$ defined as follows:

$$D_f(x, y) := f(x) - f(y) - \langle \nabla f(y), x - y \rangle. \tag{10}$$

Note that if $y = x_*$, where $x_*$ is a minimum of $f$, then $D_f(x, x_*) = f(x) - f(x_*)$.

Lastly, we define an object that plays the key role in our analysis.

**Definition 1** (Variance at optimum). *Gradient variance at optimum is the quantity*

$$\sigma_*^2 := \frac{1}{n} \sum_{i=1}^{n} \|\nabla f_i(x_*)\|^2. \tag{11}$$

This quantity is used in several recent papers on stochastic gradient-type methods. Particularly, it is a version of gradient noise introduced in Gower et al. (2019) for finite sum problems.

For all theorems in this paper the following assumption is used.

**Assumption 1.** *The objective $f$ and the individual losses $f_1, \ldots, f_n$ are all $L$-smooth. We also assume the existence of a minimizer $x_* \in \mathbb{R}^d$.*

This assumption is classical in the literature, and it is necessary for us to get convergence results for all the methods described above.

### 3.2 Convergence Analysis of Rand-Shuffle and Rand-Reshuffle

We provide two different rates in the strongly convex case. Let $\kappa := L/\mu$.

**Theorem 1** (Strongly convex case: $f$). *Suppose that each $f_i$ is convex, $f$ is $\mu$-strongly convex, and Assumption 1 holds. If the stepsize satisfies $0 < \gamma \le (2\sqrt{2}Ln\sqrt{\kappa})^{-1}$, the iterates generated by* Rand-Shuffle *and* Rand-Reshuffle *satisfy*

$$\mathbb{E}\left[\|x_T - x_*\|^2\right] \le \left(1 - \frac{\gamma n \mu}{2}\right)^T \|x_0 - x_*\|^2.$$

*This means that the iteration complexity of these methods is $T = O\left(\kappa\sqrt{\kappa}\log 1/\varepsilon\right)$.*

If we are in the big data regime characterized by the inequality $n > O(\kappa)$, then we can use a larger step-size, which leads to an improved rate. This is captured by our next theorem.

**Theorem 2** (Strongly convex case: $f$). *Suppose that each $f_i$ is convex, $f$ is $\mu$-strongly convex and Assumption 1 holds. Additionally assume we are in the "big data" regime characterized by $n \ge 2\kappa/(1 - \frac{1}{\sqrt{2}\kappa})$. Then provided the stepsize satisfies $\gamma \le 1/\sqrt{2}Ln$, the iterates generated by* Rand-Shuffle *and* Rand-Reshuffle *satisfy*

$$\mathbb{E}\left[\|x_T - x_*\|^2\right] \le \left(1 - \frac{\gamma n \mu}{2}\right)^T \|x_0 - x_*\|^2.$$

*This means that the iteration complexity of these methods is $T = O\left(\kappa\log 1/\varepsilon\right)$.*

As we shall see next, we obtain an even better rate in the case when each function $f_i$ is strongly convex.

**Theorem 3** (Strongly convex case: $f_i$). *Suppose that the functions $f_1, \ldots, f_n$ are $\mu$-strongly convex and Assumption 1 holds. Fix constant $0 < \delta < 1$. If the stepsize satisfies $\gamma \leq \delta/L\sqrt{2n\kappa}$, and if number of functions is sufficiently big, $n > \log(1-\delta^2)/\log(1-\gamma\mu)$, then the iterates generated by* Rand-Shuffle *and* Rand-Reshuffle *satisfy*

$$\mathbb{E}\left[\|x_T - x_*\|^2\right] \leq \left((1-\gamma\mu)^n + \delta^2\right)^T \|x_0 - x_*\|^2.$$

*If we further assume that $\delta^2 \leq (1-\gamma\mu)^{n/2}\left(1 - (1-\gamma\mu)^{n/2}\right)$, then the iteration complexity of these methods is $T = O\left(\kappa\sqrt{\kappa/n}\log 1/\varepsilon\right)$.*

In our work we provide the first bounds for SVRG under random reshuffling without strong convexity.

**Theorem 4** (Convex case). *Suppose the functions $f_1, f_2, \ldots, f_n$ are convex and Assumption 1 holds. Then for* Rand-Shuffle *and* Rand-Reshuffle *with stepsize $\gamma \leq 1/\sqrt{2}Ln$, the average iterate $\hat{x}_T := \frac{1}{T}\sum_{t=1}^{T} x_t$ satisfies*

$$\mathbb{E}\left[f(\hat{x}_T)\right] - f(x_*) \leq \frac{3\|x_0 - x_*\|^2}{2\gamma nT}.$$

*This means that the iteration complexity of these methods is $T = O\left(\frac{L\|x_0 - x_*\|^2}{\varepsilon}\right)$.*

We also obtained first convergence result for Rand-Reshuffle in the non-convex case.

**Theorem 5** (General non-convex case). *Suppose that Assumption 1 holds. Then for Algorithm* Rand-Reshuffle *run for $T$ epochs with a stepsize $\gamma \leq \frac{1}{2Ln}$ we have*

$$\frac{1}{T}\sum_{t=0}^{T-1}\mathbb{E}\left[\|\nabla f(x_t)\|^2\right] \leq \frac{4(f(x_0) - f_*)}{\gamma nT}.$$

*Choose $\gamma = \frac{1}{2nL}$. Then the mean of gradient norms satisfies $\frac{1}{T}\sum_{t=0}^{T-1}\mathbb{E}\left[\|\nabla f(x_t)\|^2\right] \leq \varepsilon^2$ provided the number of iterations satisfies $T = O\left(\frac{8\delta_0 L}{\varepsilon^2}\right)$.*

**Theorem 6** (Polyak-Łojasiewicz condition). *Suppose that Assumption 1 holds and $f$ satisfies the Polyak-Łojasiewicz inequality with $\mu > 0$, i.e., $\|\nabla f(x)\|^2 \geq 2\mu(f(x) - f_*)$ for any $x \in \mathbb{R}^d$. Then for Algorithm* Rand-Reshuffle *run for $T$ epochs with a stepsize $\gamma \leq \frac{1}{2Ln}$ we have*

$$\mathbb{E}\left[f(x_T) - f_*\right] \leq \left(1 - \frac{\gamma\mu n}{2}\right)^T (f(x_0) - f_*),$$

*then the relative error satisfies $\frac{\mathbb{E}[f(x_T) - f_*]}{f(x_0) - f_*} \leq \varepsilon$ provided the number of iterations satisfies $T = O(\kappa\log\frac{1}{\varepsilon})$.*

### 3.3 Convergence Analysis of Det-Shuffle

In this section we present results for Det-Shuffle. They are very similar to the previous bounds. However, the lack of randomization does not allow us to improve convergence in the big data regime.

**Theorem 7** (Strongly convex case: $f$). *Suppose that each $f_i$ is convex function, $f$ is $\mu$-strongly convex function, and Assumption 1 holds. If the stepsize satisfies $\gamma \leq 1/4Ln\sqrt{\kappa}$, the iterates generated by* Det-Shuffle *satisfy*

$$\|x_T - x_*\|^2 \leq \left(1 - \frac{\gamma n\mu}{2}\right)^T \|x_0 - x_*\|^2.$$

*This means that the iteration complexity of this method is $T = O\left(\kappa\sqrt{\kappa}\log 1/\varepsilon\right)$.*

Note that this is the same rate as that of Rand-Shuffle and Rand-Reshuffle.

Our rate for Det-Shuffle is better than the rate of Cyclic-SAGA (Park & Ryu, 2020). We remark that the convergence rate of DIAG (Mokhtari et al., 2018) is better still; however, their result requires strong convexity of each $f_i$.

Similarly, we can establish convergence results for Det-Shuffle in the convex case.

**Theorem 8** (Convex case). *Suppose the functions $f_1, f_2, \ldots, f_n$ are convex and Assumption 1 holds. If the stepsize satisfies $\gamma \leq 1/2\sqrt{2}Ln$, the average iterate $\hat{x}_T := \frac{1}{T}\sum_{j=1}^{T} x_j$ generated by* Det-Shuffle *satisfies*

$$\mathbb{E}\left[f(\hat{x}_T)\right] - f(x_*) \leq \frac{2\|x_0 - x_*\|^2}{\gamma nT}.$$

*This means that the iteration complexity of this method is $T = O\left(\frac{L\|x_0 - x_*\|^2}{\varepsilon}\right)$.*

Up to a constant factor, the complexity of Det-Shuffle is the same as that of Rand-Shuffle and Rand-Reshuffle.

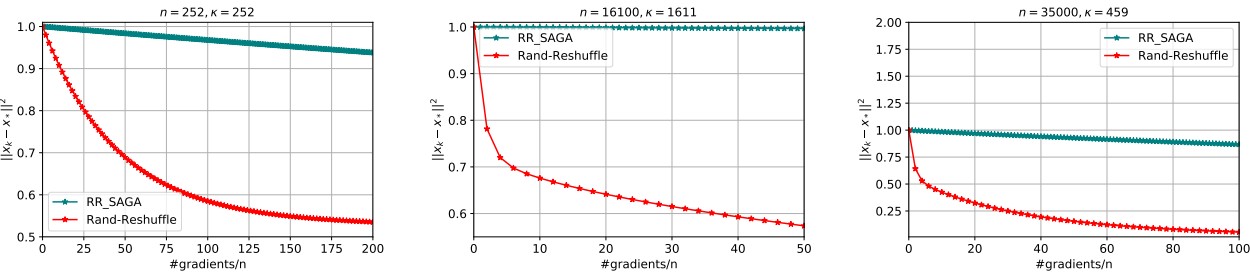

Figure 1: Comparison of Rand-Reshuffle and RR-SAGA with theoretical stepsizes on `bodyfat`, `a7a`, and `ijcnn1` datasets (from left to right).

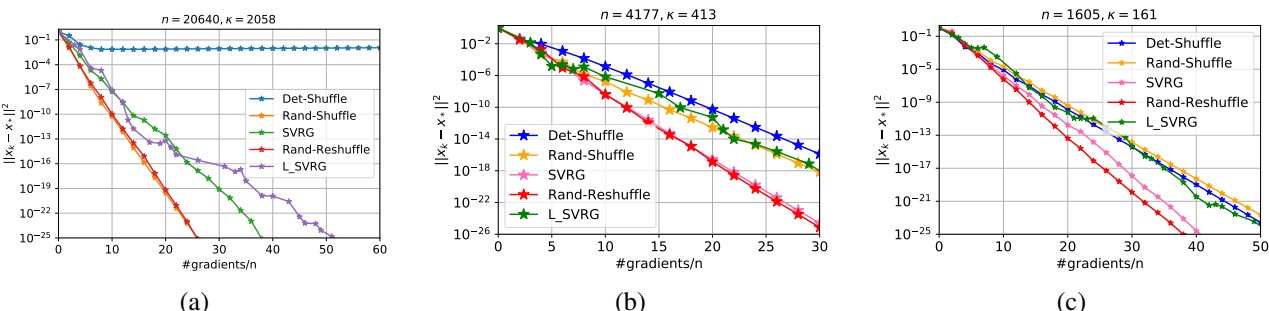

Figure 2: (a) Comparison of methods on `cadata` dataset, we set the regularization constant $\lambda = 10/n$ and carefully chosen stepsizes. (b, c) Comparison of SVRG, L-SVRG, Rand-Reshuffle, Det-Shuffle and Rand-Shuffle on `abalone` and `a1a` datasets. For each dataset we run 5 experiments and use average errors for each algorithm.

## 4 Experiments

In our experiments we solve the regularized ridge regression problem, which has the form equation 1 with

$$f_i(x) = \tfrac{1}{2}\|A_{i,:}x - y_i\|^2 + \tfrac{\lambda}{2}\|x\|^2,$$

where $A \in \mathbb{R}^{n \times d}, y \in \mathbb{R}^n$ and $\lambda > 0$ is a regularization parameter. Note that this problem is strongly convex and satisfies the Assumptions 1 for $L = \max_i \|A_{i,:}\|^2 + \lambda$ and $\mu = \lambda_{\min}(A^\top A)/n + \lambda$, where $\lambda_{\min}$ is the smallest eigenvalue. To have a tighter bound on the $L$-smoothness constant we normalize rows of the data matrix $A$. We use datasets from open LIBSVM corpus (Chang & Lin, 2011). In the plots $x$-axis is the number of single data gradient computation divided by $n$, and $y$-axis is the normalized error of the argument $\|x_k - x_*\|^2/\|x_0 - x_*\|^2$. In the appendix you can find the details and additional experiments.

### 4.1 Rand-Reshuffle vs RR-SAGA

In this experiment, we compare Rand-Reshuffle and RR-SAGA under an academic setting, i.e. we choose the steps that are suggested by theory. For Rand-Reshuffle we take the stepsize $\gamma = 1/\sqrt{2}Ln$ when $n \geq \frac{2L}{\mu} \frac{1}{1 - \frac{\mu}{\sqrt{2}L}}$ and $\gamma = \frac{1}{2\sqrt{2}Ln} \sqrt{\frac{\mu}{L}}$ otherwise, and for RR-SAGA $\gamma = \frac{\mu}{11L^2 n}$. We can see that Rand-Reshuffle outperforms RR-SAGA in terms of the number of epochs and the number of gradient computations. Although the cost of iteration of Rand-Reshuffle is twice higher than RR-SAGA, the larger stepsize significantly impacts the total complexity. In addition, RR-SAGA needs $O(nd)$ extra storage to maintain the table of gradients, which makes RR-SAGA algorithm hard to use in the big data regime.

### 4.2 Variance Reduced Random Reshuffling Algorithms

This section compares the variance reduced algorithms with and without random reshuffling: SAGA, RR-SAGA, SVRG, L-SVRG and Rand-Reshuffle. For each algorithm, we choose its optimal stepsizes using the grid search. To make algorithms

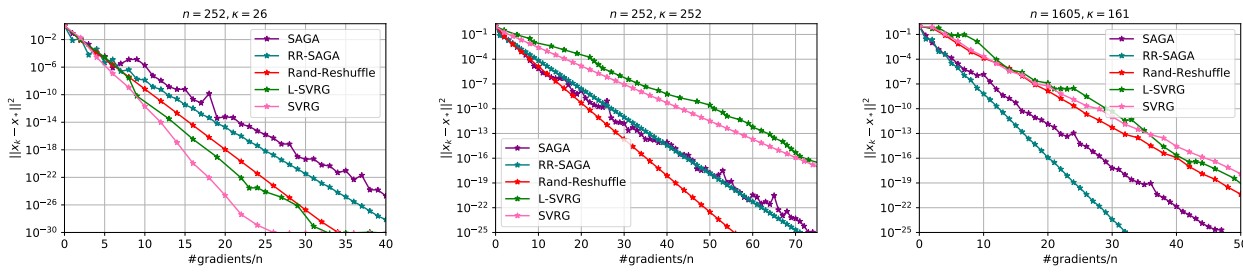

Figure 3: Comparison of SAGA, RR-SAGA, Rand-Reshuffle, L-SVRG and SVRG with optimal stepsizes on bodyfat dataset with different regularization constants (on the left and middle) and a1a (on the right).

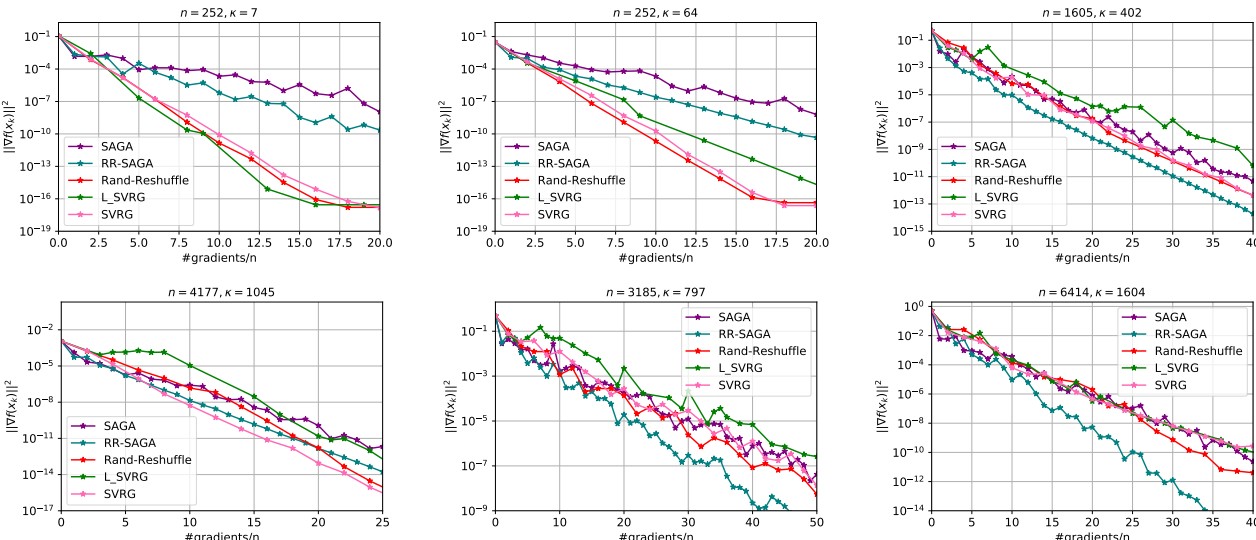

Figure 4: Comparison of SAGA, RR-SAGA, Rand-Reshuffle, L-SVRG and SVRG with optimal stepsizes on bodyfat dataset with different regularization constants (upper left and middle), a1a (upper right), abalone (lower left), a3a (lower middle) and a5a (lower right).

reasonable to compare in SVRG, we set the length of the inner loop $m = n$, in L-SVRG the control update probability is $1/n$. Also, we consider only the uniform sampling version of SVRG and L-SVRG. We can see the results on Figure 3. We can see that the variance reduced algorithms perform well on this experiment, and there is no obvious leader. However, note that for SAGA and RR-SAGA, we need to have an additional $O(nd)$ space to store the table of the gradients, which is a serious issue in the big data regime.

### 4.3 Different versions of SVRG

In this section, we compare different types of SVRG algorithm: SVRG, L-SVRG, Rand-Reshuffle, Rand-Shuffle and Det-Shuffle. For each algorithm we run five experiments with different random seeds with optimal stepsizes found by grid search. We can see that Rand-Reshuffle in average outperforms other algorithms, while in some random cases L-SVRG can perform better. Also, we can see that Rand-Shuffle is better than Det-Shuffle that coincides with theoretical findings. If the sampling in each epoch is problematic, one can shuffle data once before the training.

### 4.4 Experiments with logistic regression

We also run experiments for the regularized logistic regression problem; i.e., for problem equation 1 with

$$f(x) = \frac{1}{n} \sum_{i=1}^{n} \log\left(1 + \exp(-y_i a_i^\top x)\right) + \frac{\lambda}{2}\|x\|^2.$$

Note that the problem is $L$-smooth and $\mu$-strongly convex for $L = \frac{1}{4n}\lambda_{\max}(A^\top A) + \lambda$, and $\mu = \lambda$. In these experiments (also in the ridge regression experiments) when we choose optimal stepsize, we choose the best one among $\{\frac{1}{L}, \frac{1}{2L}, \frac{1}{3L}, \frac{1}{5L}, \frac{1}{10L}\}$. For the logistic regression we do not have an explicit formula for the optimum $x_*$ as in the ridge regression, thus in this case we compare the norm of the gradients instead. In Figure 4 we can see the performance of the variance reduced algorithms: SAGA, RR-SAGA, SVRG, L-SVRG and Rand-Reshuffle.

## 5 Conclusion

In this paper, we consider variance-reduced algorithms under random reshuffling. Our results are predominantly theoretical because these algorithms are already widely used in practice and show excellent work. We have proposed a new approach for analysis using inner product reformulation, which leads to better rates. Experimental results confirm our theoretical discoveries. Thus, we receive a deeper theoretical understanding of these algorithms' work, and we hope that this will inspire researchers to develop further and analyze these methods. The understanding of variance reduction mechanism is essential to construct accelerated versions for stochastic algorithms. We also believe that our theoretical results can be applied to other aspects of machine learning, leading to improvements in state of the art for current or future applications.

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

# A  Appendix

## Contents

# B  Basic Facts

## B.1  Elementary Inequalities

**Proposition 2.** *For all $a, b \in \mathbb{R}^d$ and $t > 0$ the following inequalities hold*

$$\langle a, b \rangle \leq \frac{\|a\|^2}{2t} + \frac{t\|b\|^2}{2},$$
$$\|a + b\|^2 \leq 2\|a\|^2 + 2\|b\|^2, \tag{12}$$
$$\frac{1}{2}\|a\|^2 - \|b\|^2 \leq \|a + b\|^2.$$

## B.2  Convexity and smoothness

**Proposition 3.** *Let $f : \mathbb{R}^d \to \mathbb{R}$ be continuously differentiable and let $L \geq 0$. Then the following statements are equivalent:*

- *$f$ is L-smooth,*

- *$2D_f(x, y) \leq L\|x - y\|^2$ for all $x, y \in \mathbb{R}^d$,*

- *$\langle \nabla f(x) - \nabla f(y), x - y \rangle \leq L\|x - y\|^2$ for all $x, y \in \mathbb{R}^d$.*

**Proposition 4.** *Let $f : \mathbb{R}^d \to \mathbb{R}$ be continuously differentiable and let $\mu \geq 0$. Then the following statements are equivalent:*

- *$f$ is $\mu$-strongly convex,*

- *$2D_f(x, y) \geq \mu\|x - y\|^2$ for all $x, y \in \mathbb{R}^d$,*

- *$\langle \nabla f(x) - \nabla f(y), x - y \rangle \geq \mu\|x - y\|^2$ for all $x, y \in \mathbb{R}^d$.*

Note that the $\mu = 0$ case reduces to convexity.

**Proposition 5.** *Let $f : \mathbb{R}^d \to \mathbb{R}$ be continuously differentiable and $L > 0$. Then the following statements are equivalent:*

- *$f$ is convex and L-smooth*

- *$0 \leq 2D_f(x, y) \leq L\|x - y\|^2$ for all $x, y \in \mathbb{R}^d$,*

- *$\frac{1}{L}\|\nabla f(x) - \nabla f(y)\|^2 \leq 2D_f(x, y)$ for all $x, y \in \mathbb{R}^d$,*

- *$\frac{1}{L}\|\nabla f(x) - \nabla f(y)\|^2 \leq \langle \nabla f(x) - \nabla f(y), x - y \rangle$ for all $x, y \in \mathbb{R}^d$.*

**Proposition 6** (Jensen's inequality). *Let $f : \mathbb{R}^d \to \mathbb{R}$ be a convex function, $x_1, \ldots, x_m \in \mathbb{R}^d$, and $\lambda_1, \ldots, \lambda_m$ be nonnegative real numbers adding up to 1. Then*

$$f\left(\sum_{i=1}^{m} \lambda_i x_i\right) \leq \sum_{i=1}^{m} \lambda_i f(x_i).$$

## B.3  From convergence rate to iteration complexity

We implicitly use the following standard result to derive iteration complexity results in our theorems. We include the statement and proof, for completeness.

**Lemma 2.** *Consider a randomized algorithm producing a sequence of random iterates $\{x_t\}_{t \geq 0}$. Let $D_t$ be some nonnegative function of $x_t$ (example: $D_t = \|x_t - x_*\|^2$). Assume that there exists $q \in (0, 1)$ such that the following inequality holds for all $t \geq 0$:*

$$\mathbb{E}[D_t] \leq (1 - q)^t D_0. \tag{13}$$

*Fix any $\varepsilon > 0$. Then as long as*

$$T \geq \frac{1}{q} \ln\left(\frac{1}{\varepsilon}\right),$$

*we have*

$$\mathbb{E}[D_T] \leq \varepsilon D_0.$$

*Proof.* Since $e^q \geq 1 + q$ for all $q \in \mathbb{R}$, we have $e^{-q} \geq 1 - q$ for all $q \in (0, 1)$. Since logarithm is an increasing over $\mathbb{R}_+$, it follows that $-q \geq \ln(1 - q)$ for all $q \in (0, 1)$. Therefore, the inequality

$$-tq \geq t \ln(1 - q)$$

holds for all $t \geq 0$ and all $q \in (0, 1)$. Now if we have $T \geq \frac{1}{q} \ln\left(\frac{1}{\varepsilon}\right)$, which is equivalent to $-T \cdot q \leq \ln(\varepsilon)$, we obtain $T \ln(1 - q) \leq \ln(\varepsilon)$. Taking exponential on both sides, we get

$$0 < (1 - q)^T \leq \varepsilon. \tag{14}$$

Finally, we have

$$\mathbb{E}[D_T] \overset{\text{equation 13}}{\leq} (1 - q)^T D_0 \overset{\text{equation 14}}{\leq} \varepsilon D_0.$$

$\square$

## C   Proof of Proposition 1

Assume that each $f_i$ is $\mu$-strongly convex (resp. convex) and $L$-smooth. Then the function

$$f^t := \frac{1}{n} \sum_{i=1}^{n} f_i^t,$$

where

$$f_i^t(x) := f_i(x) + \left\langle a_i^t, x \right\rangle, \tag{15}$$

is $\mu$-strongly convex (resp. convex) and $L$-smooth.

*Proof.* Let us compute Bregman divergence with respect to the new function $f_i^t(x)$ :

$$D_{f_i^t}(x, y) = f_i^t(x) - f_i^t(y) - \langle \nabla f_i^t(y), x - y \rangle.$$

Note that $\nabla f_i^t(y) = \nabla f_i(y) + a_i^t$. Now we have

$$
\begin{aligned}
D_{f_i^t}(x, y) &= f_i^t(x) - f_i^t(y) - \langle \nabla f_i^t(y), x - y \rangle \\
&= f_i(x) + \left\langle a_i^t, x \right\rangle - \left( f_i(y) + \left\langle a_i^t, y \right\rangle \right) - \langle \nabla f_i(y) + a_i^t, x - y \rangle \\
&= f_i(x) + \left\langle a_i^t, x \right\rangle - f_i(y) - \left\langle a_i^t, y \right\rangle - \langle \nabla f_i(y), x - y \rangle - \langle a_i^t, x - y \rangle \\
&= f_i(x) + \left\langle a_i^t, x \right\rangle - f_i(y) - \left\langle a_i^t, y \right\rangle - \langle \nabla f_i(y), x - y \rangle - \langle a_i^t, x \rangle + \langle a_i^t, y \rangle \\
&= f_i(x) - f_i(y) - \langle \nabla f_i(y), x - y \rangle \\
&= D_{f_i}(x, y).
\end{aligned}
$$

Since the Bregman divergence is not changed, the new function $f_i^t(x)$ has the same properties ($\mu$-strong convexity or convexity and $L$-smoothness) as the initial function $f_i(x)$.   □

# D Proof of Lemma 1

*Proof.* Using the fact that $\nabla f(x_*) = 0$, applying Young's inequality equation 12, and finally employing several standard inequalities from Section B.2, we get

$$
\begin{aligned}
\left(\sigma_*^t\right)^2 \quad &:= \quad \frac{1}{n} \sum_{i=1}^{n} \|\nabla f_i^t(x_*)\|^2 \\
&= \quad \frac{1}{n} \sum_{i=1}^{n} \|\nabla f_i(x_*) - \nabla f_i(y_t) + \nabla f(y_t)\|^2 \\
&= \quad \frac{1}{n} \sum_{i=1}^{n} \|\nabla f_i(x_*) - \nabla f_i(y_t) + \nabla f(y_t) - \nabla f(x_*)\|^2 \\
&\overset{equation\ 12}{\leq} \quad \frac{1}{n} \sum_{i=1}^{n} \left(2\|\nabla f_i(y_t) - \nabla f_i(x_*)\|^2 + 2\|\nabla f(y_t) - \nabla f(x_*)\|^2\right) \\
&\leq \quad \frac{1}{n} \sum_{i=1}^{n} 4L_i D_{f_i}(y_t, x_*) + \frac{1}{n} \sum_{i=1}^{n} 4L D_f(y_t, x_*) \\
&\leq \quad 4L D_f(y_t, x_*) + 4L D_f(y_t, x_*) \\
&= \quad 8L D_f(y_t, x_*) \\
&\leq \quad 4L^2 \|y_t - x_*\|^2.
\end{aligned}
$$

$\square$

# E  Analysis of Rand-Shuffle and Rand-Reshuffle

## E.1  Proof of Theorems 1 and 2

*Proof.* We start from Lemma 3 in paper of Mishchenko et al. (2020).

**Lemma 3.** Assume that functions $f_1, \ldots, f_n$ are convex and that Assumption 1 is satisfied. If Random Reshuffling or Shuffle-Once is run with a stepsize satisfying $\gamma \leq \frac{1}{\sqrt{2}Ln}$, then

$$\mathbb{E}\left[\|x_{t+1} - x_*\|^2\right] \leq \mathbb{E}\left[\|x_t - x_*\|^2\right] - 2\gamma n\mathbb{E}\left[f(x_{t+1}) - f(x_*)\right] + \frac{\gamma^3 Ln^2\sigma_*^2}{2}.$$

Now we can apply this inequality to the reformulated problem equation 2. Using strong convexity, we obtain

$$\mathbb{E}\left[\|x_{t+1} - x_*\|^2 \mid x_t\right] \leq \|x_t - x_*\|^2 - 2\gamma n\mathbb{E}\left[f(x_{t+1}) - f(x_*) \mid x_t\right] + \frac{\gamma^3 Ln^2\left(\sigma_*^t\right)^2}{2}$$

$$\leq \|x_t - x_*\|^2 - \gamma n\mu\mathbb{E}\left[\|x_{t+1} - x_*\|^2 \mid x_t\right] + \frac{\gamma^3 Ln^2\left(\sigma_*^t\right)^2}{2}.$$

Since we update $y_t = x_t$ after each epoch, this leads to

$$\mathbb{E}\left[\|x_{t+1} - x_*\|^2 \mid x_t\right] \leq \frac{1}{1 + \gamma\mu n}\left(\|x_t - x_*\|^2 + \frac{\gamma^3 Ln^2\left(\sigma_*^t\right)^2}{2}\right)$$

$$\overset{\text{equation 6}}{\leq} \frac{1}{1 + \gamma\mu n}\left(\|x_t - x_*\|^2 + \frac{\gamma^3 Ln^2 \cdot 4L^2\|y_t - x_*\|^2}{2}\right)$$

$$= \frac{1}{1 + \gamma\mu n}\left(\|x_t - x_*\|^2 + 2\gamma^3 n^2 L^3\|x_t - x_*\|^2\right)$$

$$= \frac{1}{1 + \gamma\mu n}\left(1 + 2\gamma^3 n^2 L^3\right)\|x_t - x_*\|^2.$$

We can use the tower property to obtain

$$\mathbb{E}\left[\|x_{t+1} - x_*\|^2\right] \leq \frac{1 + 2\gamma^3 L^3 n^2}{1 + \gamma\mu n}\mathbb{E}\left[\|x_t - x_*\|^2\right].$$

If this inequality $\frac{1+2\gamma^3 L^3 n^2}{1+\gamma\mu n} \leq 1 - \frac{\gamma n\mu}{2}$ is correct, we can unroll the recursion and obtain

$$\mathbb{E}\left[\|x_T - x_*\|^2\right] \leq \left(1 - \frac{\gamma n\mu}{2}\right)^T \|x_0 - x_*\|^2.$$

Now we need to solve the following inequality:

$$\frac{1 + 2\gamma^3 L^3 n^2}{1 + \gamma\mu n} \leq 1 - \frac{\gamma n\mu}{2}.$$

Let us simplify it:

$$1 + 2\gamma^3 L^3 n^2 \leq 1 + \frac{\gamma n\mu}{2} - \frac{\gamma^2 n^2\mu^2}{2}$$

$$2\gamma^2 L^3 n^2 \leq \frac{n\mu}{2} - \frac{\gamma n^2\mu^2}{2}$$

$$2\gamma^2 L^3 n \leq \frac{\mu}{2} - \frac{\gamma n\mu^2}{2}$$

$$2\gamma^2 L^3 n + \frac{\gamma n\mu^2}{2} \leq \frac{\mu}{2}.$$

Now as $\gamma \le \frac{1}{2\sqrt{2}Ln}\sqrt{\frac{\mu}{L}}$, we have

$$2 \cdot \frac{1}{8L^2n^2} \cdot \frac{\mu}{L}L^3n + \frac{1}{2\sqrt{2}Ln}\sqrt{\frac{\mu}{L}} \cdot \frac{n\mu^2}{2} \le \frac{\mu}{2}$$

$$\frac{1}{4n}\mu + \frac{1}{4\sqrt{2}}\frac{\mu}{L}\sqrt{\frac{\mu}{L}}\mu \le \frac{\mu}{2}$$

$$\frac{1}{4n} + \frac{1}{4\sqrt{2}}\frac{\mu}{L}\sqrt{\frac{\mu}{L}} \le \frac{1}{2}.$$

It is true since $n \ge 1$ and $\mu \le L$. We have proved Theorem 1.

Now let us use the biggest step-size allowed by the Lemma 3 in Section E.1. Let us utilize $\gamma \le \frac{1}{\sqrt{2}Ln}$ :

$$2 \cdot \frac{1}{2L^2n^2}L^3n + \frac{1}{\sqrt{2}Ln} \cdot \frac{n\mu^2}{2} \le \frac{\mu}{2}$$

$$\frac{L}{n} + \frac{\mu}{2} \cdot \frac{\mu}{\sqrt{2}L} \le \frac{\mu}{2}.$$

This leads to

$$\frac{L}{n} \le \frac{\mu}{2} - \frac{\mu}{2} \cdot \frac{\mu}{\sqrt{2}L} = \frac{\mu}{2}\left(1 - \frac{\mu}{\sqrt{2}L}\right)$$

and

$$\frac{1}{n} \le \frac{\mu}{2L}\left(1 - \frac{\mu}{\sqrt{2}L}\right) \quad \Rightarrow \quad n \ge \frac{2L}{\mu} \cdot \frac{1}{1 - \frac{\mu}{\sqrt{2}L}}.$$

We have proved Theorem 2. $\qquad\qquad\qquad\qquad\qquad\qquad\qquad\qquad\qquad\qquad\qquad$ □

## E.2   Proof of Theorem 3

We start from Theorem 1 in (Mishchenko et al., 2020), which states that

$$\mathbb{E}\left[\|x_{t+1} - x_*\|^2 \mid x_t\right] \le (1 - \gamma\mu)^n \|x_t - x_*\|^2 + 2\gamma^2\sigma^2_{\text{Shuffle}}\left(\sum_{i=0}^{n-1}(1 - \gamma\mu)^i\right).$$

Using Proposition 1 from (Mishchenko et al., 2020), which says that

$$\frac{\gamma\mu n}{8}\sigma^2_* \le \sigma^2_{\text{Shuffle}} \le \frac{\gamma Ln}{4}\sigma^2_*,$$

we get

$$\mathbb{E}\left[\|x_{t+1} - x_*\|^2 \mid x_t\right] \le (1 - \gamma\mu)^n \|x_t - x_*\|^2 + \frac{\gamma^3 Ln}{2}\sigma^2_*\left(\sum_{i=0}^{n-1}(1 - \gamma\mu)^i\right)$$

$$\le (1 - \gamma\mu)^n \|x_t - x_*\|^2 + \frac{\gamma^2 Ln}{2\mu}\sigma^2_*.$$

Now we can apply Lemma 1 and using $y_t = x_t$ we have the following inequality:

$$\mathbb{E}\left[\|x_{t+1} - x_*\|^2 \mid x_t\right] \le (1 - \gamma\mu)^n \|x_t - x_*\|^2 + \frac{2\gamma^2 L^3 n}{\mu}\|x_t - x_*\|^2$$

$$\le \left((1 - \gamma\mu)^n + \frac{2\gamma^2 L^3 n}{\mu}\right)\|x_t - x_*\|^2.$$

Applying the tower property, we get

$$\mathbb{E}\left[\|x_{t+1} - x_*\|^2\right] \leq \left((1 - \gamma\mu)^n + \frac{2\gamma^2 L^3 n}{\mu}\right)\mathbb{E}\left[\|x_t - x_*\|^2\right],$$

and after unrolling this recursion, we get

$$\mathbb{E}\left[\|x_T - x_*\|^2\right] \leq \left((1 - \gamma\mu)^n + \frac{2\gamma^2 L^3 n}{\mu}\right)^T \mathbb{E}\left[\|x_0 - x_*\|^2\right]$$

$$\leq \left((1 - \gamma\mu)^n + \frac{\delta^2}{L^2}\frac{\mu}{2nL}\frac{2L^3 n}{\mu}\right)^T \mathbb{E}\left[\|x_0 - x_*\|^2\right]$$

$$\leq \left((1 - \gamma\mu)^n + \delta^2\right)^T \mathbb{E}\left[\|x_0 - x_*\|^2\right],$$

where we used the stepsize restriction $\gamma \leq \frac{\delta}{L}\sqrt{\frac{\mu}{2nL}}$. In order for this to lead to convergence, we need to assume that $(1 - \gamma\mu)^n + \delta^2 < 1$. This is satisfied, for example, if $n$ is large enough. In particular, this holds when

$$n > \log\left(\frac{1}{1 - \delta^2}\right) \cdot \left(\log\left(\frac{1}{1 - \gamma\mu}\right)\right)^{-1}.$$

Finally, using the additional assumption $\delta^2 \leq (1 - \gamma\mu)^{\frac{n}{2}}\left(1 - (1 - \gamma\mu)^{\frac{n}{2}}\right)$, we get

$$\delta^2 + (1 - \gamma\mu)^n \leq (1 - \gamma\mu)^{\frac{n}{2}}.$$

Now we can apply Theorem 3 and get

$$\mathbb{E}\left[\|x_T - x_*\|^2\right] \leq (1 - \gamma\mu)^{\frac{nT}{2}}\|x_0 - x_*\|^2.$$

Finally, we apply Lemma 2 with $\gamma = \frac{\delta}{L}\sqrt{\frac{\mu}{2nL}}$ and get iteration complexity $T = O\left(\kappa\sqrt{\frac{\kappa}{n}}\log\left(\frac{1}{\varepsilon}\right)\right)$.

### E.3 Proof of Theorem 4

Suppose the functions $f_1, f_2, \ldots, f_n$ are convex and Assumption 1 holds. Then for Rand-Reshuffle or Rand-Shuffle with stepsize $\gamma \leq \frac{1}{\sqrt{2}Ln}$, the average iterate $\hat{x}_T := \frac{1}{T}\sum_{t=1}^{T} x_t$ satisfies

$$\mathbb{E}\left[f(\hat{x}_T) - f(x_*)\right] \leq \frac{3\|x_0 - x_*\|^2}{2\gamma nT}.$$

*Proof.* We start with Lemma 3 from Mishchenko et al. (2020), which says that

$$\mathbb{E}\left[\|x_{t+1} - x_*\|^2 \mid x_t\right] \leq \|x_t - x_*\|^2 - 2\gamma n\mathbb{E}\left[f(x_{t+1}) - f(x_*) \mid x_t\right] + \frac{\gamma^3 Ln^2\sigma_*^2}{2}.$$

Apply this inequality to the reformulated problem equation 2, we get

$$2\gamma n\mathbb{E}\left[f(x_{t+1}) - f(x_*) \mid x_t\right] \leq \|x_t - x_*\|^2 - \mathbb{E}\left[\|x_{t+1} - x_*\|^2 \mid x_t\right] + \frac{\gamma^3 Ln^2(\sigma_*^t)^2}{2}. \tag{16}$$

Using Lemma 1 and the fact that $y_t = x_t$ and $f = f^t$, we get

$$\left(\sigma_*^t\right)^2 \leq 8LD_{f^t}(x_t, x_*) = 8LD_f(x_t, x_*) = 8L(f(x_t) - f(x_*)), \tag{17}$$

where the last identity follows from Proposition 1.

Plugging equation 17 into equation 16, we obtain

$$2\gamma n \mathbb{E}\left[f\left(x_{t+1}\right) - f\left(x_*\right) \mid x_t\right] \le \|x_t - x_*\|^2 - \mathbb{E}\left[\|x_{t+1} - x_*\|^2 \mid x_t\right] + \frac{\gamma^3 L n^2}{2} \cdot 8L(f(x_t) - f(x_*)),$$

which after using the tower property turns into

$$2\gamma n \mathbb{E}\left[f\left(x_{t+1}\right) - f\left(x_*\right)\right] \le \mathbb{E}\left[\|x_t - x_*\|^2\right] - \mathbb{E}\left[\|x_{t+1} - x_*\|^2\right] + 4\gamma^3 L^2 n^2 \mathbb{E}\left[f(x_t) - f(x_*)\right].$$

Now we subtract from both sides:

$$2\gamma n \mathbb{E}\left[f\left(x_{t+1}\right) - f\left(x_*\right)\right] - 4\gamma^3 L^2 n^2 \mathbb{E}\left[f\left(x_{t+1}\right) - f\left(x_*\right)\right] \le \mathbb{E}\left[\|x_t - x_*\|^2\right] - \mathbb{E}\left[\|x_{t+1} - x_*\|^2\right]$$
$$+ 4\gamma^3 L^2 n^2 \mathbb{E}\left[f(x_t) - f(x_*)\right]$$
$$- 4\gamma^3 L^2 n^2 \mathbb{E}\left[f\left(x_{t+1}\right) - f\left(x_*\right)\right]$$
$$\left(2\gamma n - 4\gamma^3 L^2 n^2\right) \mathbb{E}\left[f\left(x_{t+1}\right) - f\left(x_*\right)\right] \le \mathbb{E}\left[\|x_t - x_*\|^2\right] - \mathbb{E}\left[\|x_{t+1} - x_*\|^2\right]$$
$$+ 4\gamma^3 L^2 n^2 \left(\mathbb{E}\left[f(x_t) - f(x_*)\right] - \mathbb{E}\left[f\left(x_{t+1}\right) - f\left(x_*\right)\right]\right)$$
$$2\gamma n \left(1 - 2\gamma^2 L^2 n\right) \mathbb{E}\left[f\left(x_{t+1}\right) - f\left(x_*\right)\right] \le \mathbb{E}\left[\|x_t - x_*\|^2\right] - \mathbb{E}\left[\|x_{t+1} - x_*\|^2\right]$$
$$+ 4\gamma^3 L^2 n^2 \left(\mathbb{E}\left[f(x_t) - f(x_*)\right] - \mathbb{E}\left[f\left(x_{t+1}\right) - f\left(x_*\right)\right]\right).$$

Summing these inequalities for $t = 0, 1, \ldots, T - 1$ gives

$$2\gamma n \left(1 - 2\gamma^2 L^2 n\right) \sum_{t=0}^{T-1} \mathbb{E}\left[f\left(x_{t+1}\right) - f\left(x_*\right)\right] \le \sum_{t=0}^{T-1} \left(\mathbb{E}\left[\|x_t - x_*\|^2\right] - \mathbb{E}\left[\|x_{t+1} - x_*\|^2\right]\right)$$
$$+ 4\gamma^3 L^2 n^2 \sum_{t=0}^{T-1} \left(\mathbb{E}\left[f(x_t) - f(x_*)\right] - \mathbb{E}\left[f\left(x_{t+1}\right) - f\left(x_*\right)\right]\right)$$
$$= \mathbb{E}\left[\|x_0 - x_*\|^2\right] - \mathbb{E}\left[\|x_T - x_*\|^2\right]$$
$$+ 4\gamma^3 L^2 n^2 \mathbb{E}\left[f\left(x_0\right) - f\left(x_*\right)\right] - 4\gamma^3 L^2 n^2 \mathbb{E}\left[f\left(x_T\right) - f\left(x_*\right)\right]$$
$$\le \mathbb{E}\left[\|x_0 - x_*\|^2\right] + 4\gamma^3 L^2 n^2 \mathbb{E}\left[f\left(x_0\right) - f\left(x_*\right)\right]$$
$$\le \mathbb{E}\left[\|x_0 - x_*\|^2\right] + 2\gamma^3 L^3 n^2 \mathbb{E}\left[\|x_0 - x_*\|^2\right]$$
$$= (1 + 2\gamma^3 L^3 n^2) \mathbb{E}\left[\|x_0 - x_*\|^2\right],$$

and dividing both sides by $2\gamma n \left(1 - 2\gamma^2 L^2 n\right) T$, we get

$$\frac{1}{T} \sum_{t=0}^{T-1} \mathbb{E}\left[f\left(x_{t+1}\right) - f\left(x_*\right)\right] \le \frac{1 + 2\gamma^3 L^3 n^2}{1 - 2\gamma^2 L^2 n} \frac{\|x_0 - x_*\|^2}{2\gamma n T}.$$

Using the convexity of $f$, the average iterate $\hat{x}_T \overset{\text{def}}{=} \frac{1}{T} \sum_{t=1}^{T} x_t$ satisfies

$$\mathbb{E}\left[f\left(\hat{x}_T\right) - f\left(x_*\right)\right] \le \frac{1}{T} \sum_{t=1}^{T} \mathbb{E}\left[f\left(x_t\right) - f\left(x_*\right)\right] \le \frac{1 + 2\gamma^3 L^3 n^2}{1 - 2\gamma^2 L^2 n} \frac{\|x_0 - x_*\|^2}{2\gamma n T}.$$

Let us show that

$$\frac{1 + 2\gamma^3 L^3 n^2}{1 - 2\gamma^2 L^2 n} \le 3.$$

Applying $\gamma \le \frac{1}{\sqrt{2}Ln}$ we have

$$\frac{1 + 2\frac{1}{2\sqrt{2}L^3 n^3} L^3 n^2}{1 - 2\frac{1}{2L^2 n^2} L^2 n} = \frac{1 + \frac{1}{\sqrt{2}n}}{1 - \frac{1}{n}} \le 3.$$

This leads to $4n > 6 + \sqrt{2}$ and since $n \in \mathbb{N} : n > 1$, this inequality holds. Finally, we have

$$\mathbb{E}\left[f\left(\hat{x}_T\right) - f\left(x_*\right)\right] \leq \frac{3\left\|x_0 - x_*\right\|^2}{2\gamma nT}.$$

$\square$

### E.4  Proof of Theorem 5 and 6

We provide analysis for non-convex settings.

Let us remind you our reformulation:

$$f(x) = \frac{1}{n} \sum_{i=1}^{n} f_i(x) = \frac{1}{n} \sum_{i=1}^{n} \left( f_i(x) + \left\langle a_t^i, x \right\rangle \right) := \frac{1}{n} \sum_{i=1}^{n} f_i^t(x),$$

where $f_i^t(x) := f_i(x) + \left\langle a_t^i, x \right\rangle$ and $\sum_{i=1}^{n} a_t^i = 0$. Note that

$$\nabla f_i^t(x) = \nabla f_i(x) + a_t^I.$$

In particular, we choose

$$a_t^i := -\nabla f_{\pi_i}(y_t) + \nabla f(y_t).$$

Finally, we have

$$\nabla f_{\pi_i}^t(x) = \nabla f_{\pi_i}(x) - \nabla f_{\pi_i}(y_t) + \nabla f(y_t).$$

Now we need to establish an analogue of Lemma 1 for gradient variance. Let us define

$$\sigma^2(x_t) = \frac{1}{n} \sum_{i=1}^{n} \|\nabla f_i(x_t) - \nabla f(x_t)\|^2.$$

**Lemma** If we apply the linear perturbation reformulation, then the gradient variance of the reformulated problem $(\sigma_t^2)$ is equal to zero.

*Proof.*

$$\sigma_t^2(x_t) = \frac{1}{n} \sum_{i=1}^{n} \left\| \nabla f_i^t(x_t) - \nabla f(x_t) \right\|^2 = \frac{1}{n} \sum_{i=1}^{n} \left\| \nabla f_{\pi_i}(x_t) - \nabla f_{\pi_i}(y_t) + \nabla f(y_t) - \nabla f(x_t) \right\|^2.$$

In Algorithm Rand-Reshuffle we set $x_t = y_t$, we have

$$\sigma_t^2(x_t) = \frac{1}{n} \sum_{i=1}^{n} \left\| \nabla f_{\pi_i}(x_t) - \nabla f_{\pi_i}(y_t) + \nabla f(y_t) - \nabla f(x_t) \right\|^2$$

$$= \frac{1}{n} \sum_{i=1}^{n} \left\| \nabla f_{\pi_i}(x_t) - \nabla f_{\pi_i}(x_t) + \nabla f(x_t) - \nabla f(x_t) \right\|^2$$

$$= 0.$$

$\square$

Suppose that Assumption 1 holds. Then for Algorithm Rand-Reshuffle run for $T$ epochs with a stepsize $\gamma \leq \frac{1}{2Ln}$ we have

$$\frac{1}{T} \sum_{t=0}^{T-1} \mathbb{E}\left[ \|\nabla f(x_t)\|^2 \right] \leq \frac{4(f(x_0) - f_*)}{\gamma n T}.$$

Choose $\gamma = \frac{1}{2nL}$. Then the mean of gradient norms satisfies $\frac{1}{T} \sum_{t=0}^{T-1} \mathbb{E}\left[ \|\nabla f(x_t)\|^2 \right] \leq \varepsilon^2$ provided the number of iterations satisfies $T = O\left( \frac{8\delta_0 L}{\varepsilon^2} \right)$.

Suppose that Assumption 1 holds and $f$ satisfies the Polyak-Łojasiewicz inequality with $\mu > 0$, i.e., $\|\nabla f(x)\|^2 \geq 2\mu(f(x) - f_*)$ for any $x \in \mathbb{R}^d$. Then for Algorithm Rand-Reshuffle run for $T$ epochs with a stepsize $\gamma \leq \frac{1}{2Ln}$ we have

$$\mathbb{E}\left[ f(x_T) - f_* \right] \leq \left( 1 - \frac{\gamma \mu n}{2} \right)^T (f(x_0) - f_*),$$

then the relative error satisfies $\frac{\mathbb{E}[f(x_T) - f_*]}{f(x_0) - f_*} \leq \varepsilon$ provided the number of iterations satisfies $T = O(\kappa \log \frac{1}{\varepsilon})$.

*Proof.* We start from Lemma 4 and Lemma 5 from Mishchenko et al. (2020)

$$\mathbb{E}\left[f(x_{t+1})|x_t\right] \leq f(x_t) - \frac{\gamma n}{2}\|\nabla f(x_t)\|^2 + \frac{\gamma L^2}{2}\left(\gamma^2 n^3\|\nabla f(x_t)\|^2 + \gamma^2 n^2 \sigma^2(x_t)\right)$$

This lemma works for the reformulated problem. Since we do not change initial function f(x) the gradient $\nabla f(x_t)$ remains the same. The only thing that changes is the variance of the gradient. According to the lemma proved above, this variance is equal to zero. Now we have the following inequality:

$$\mathbb{E}\left[f(x_{t+1})|x_t\right] \leq f(x_t) - \frac{\gamma n}{2}\|\nabla f(x_t)\|^2 + \frac{\gamma L^2}{2}\gamma^2 n^3\|\nabla f(x_t)\|^2$$
$$\leq f(x_t) - \frac{\gamma n}{2}\left(1 - \gamma^2 L^2 n^2\right)\|\nabla f(x_t)\|^2$$

Let $\delta = f(x_t) - f_*$. Adding $-f_*$ to both sides,

$$\mathbb{E}\left[\delta_{t+1}|x_t\right] \leq \delta_t - \frac{\gamma n}{2}\left(1 - \gamma^2 L^2 n^2\right)\|\nabla f(x_t)\|^2$$

Taking unconditional expectations and using that $\gamma \leq \frac{1}{2Ln}$ we have $1 - \gamma^2 L^2 n^2 \geq \frac{1}{2}$, we get

$$\mathbb{E}\left[\delta_{t+1}\right] \leq \mathbb{E}\left[\delta_t\right] - \frac{\gamma n}{4}\mathbb{E}\left[\|\nabla f(x_t)\|^2\right].$$

It leads to

$$\frac{1}{T}\sum_{t=0}^{T-1}\mathbb{E}\left[\|\nabla f(x_t)\|^2\right] \leq \frac{4}{\gamma n}\frac{1}{T}\sum_{t=0}^{T-1}\left(\mathbb{E}\left[\delta_{t+1}\right] - \mathbb{E}\left[\delta_t\right]\right) \leq \frac{4\delta_0}{\gamma n T}$$

If we have PL condition, then we start from

$$\mathbb{E}\left[\delta_{t+1}\right] \leq \mathbb{E}\left[\delta_t\right] - \frac{\gamma n}{4}\mathbb{E}\left[\|\nabla f(x_t)\|^2\right].$$

Applying $\frac{1}{2}\|\nabla f(x)\|^2 \geq \mu(f(x) - f_*)$ leads to

$$\mathbb{E}\left[\delta_{t+1}\right] \leq \mathbb{E}\left[\delta_t\right] - \frac{\gamma \mu n}{2}\mathbb{E}\left[f(x_t) - f_*\right].$$

Unrolling this recursion, we get

$$\mathbb{E}\left[\delta_T\right] \leq \left(1 - \frac{\gamma \mu n}{2}\right)^T \delta_0.$$

Suppose that Assumption 1 holds. Choose the stepsize $\gamma$ as $\frac{1}{2nL}$. Then the mean of gradient norms satisfies

$$\frac{1}{T}\sum_{t=0}^{T-1}\mathbb{E}\left[\|\nabla f(x_t)\|^2\right] \leq \varepsilon^2$$

provided the number of iterations satisfies

$$T \geq \frac{8\delta_0 L}{\varepsilon^2}.$$

If $f$ satisfies the Polyak-Łojasiewicz inequality, then the relative error satisfies

$$\frac{\mathbb{E}\left[f(x_T) - f_*\right]}{(f(x_0) - f_*)} \leq \varepsilon$$

provided the number of iterations satisfies

$$T = O(\kappa \log \frac{1}{\varepsilon}).$$

$\square$

# F  Analysis of Det-Shuffle

## F.1  Proof of Theorem 7

We start from Lemma 8 in Mishchenko et al. (2020)

$$\|x_{t+1} - x_*\|^2 \le \|x_t - x_*\|^2 - 2\gamma n \left( f(x_{t+1}) - f(x_*) \right) + \gamma^3 L n^3 \sigma_*^2. \tag{18}$$

Now we can apply to the reformulated problem equation 2. Using strong convexity we obtain

$$\mathbb{E}\left[ \|x_{t+1} - x_*\|^2 \mid x_t \right] \le \|x_t - x_*\|^2 - 2\gamma n \mathbb{E}\left[ f(x_{t+1}) - f(x_*) \mid x_t \right] + \gamma^3 L n^2 \left( \sigma_*^t \right)^2$$

$$\le \|x_t - x_*\|^2 - \gamma n \mu \mathbb{E}\left[ \|x_{t+1} - x_*\|^2 \mid x_t \right] + \gamma^3 L n^3 \left( \sigma_*^t \right)^2.$$

Since we update $y_t = x_t$ after each epoch, this leads to

$$\mathbb{E}\left[ \|x_{t+1} - x_*\|^2 \mid x_t \right] \le \frac{1}{1 + \gamma \mu n} \left( \|x_t - x_*\|^2 + \gamma^3 L n^3 \left( \sigma_*^t \right)^2 \right)$$

$$\le \frac{1}{1 + \gamma \mu n} \left( \|x_t - x_*\|^2 + \gamma^3 L n^3 \cdot 4L^2 \|y_t - x_*\|^2 \right)$$

$$= \frac{1}{1 + \gamma \mu n} \left( \|x_t - x_*\|^2 + 4\gamma^3 n^3 L^3 \|x_t - x_*\|^2 \right)$$

$$= \frac{1}{1 + \gamma \mu n} \left( 1 + 4\gamma^3 n^3 L^3 \right) \|x_t - x_*\|^2.$$

We can use the tower property to obtain

$$\mathbb{E}\left[ \|x_{t+1} - x_*\|^2 \right] \le \frac{1 + 4\gamma^3 L^3 n^3}{1 + \gamma \mu n} \mathbb{E}\left[ \|x_t - x_*\|^2 \right].$$

If this inequality $\frac{1 + 4\gamma^3 L^3 n^3}{1 + \gamma \mu n} \le 1 - \frac{\gamma n \mu}{2}$ is correct, we can unroll the recursion and obtain

$$\mathbb{E}\left[ \|x_T - x_*\|^2 \right] \le \left( 1 - \frac{\gamma n \mu}{2} \right)^T \|x_0 - x_*\|^2.$$

Now we need to solve the following inequality:

$$\frac{1 + 4\gamma^3 L^3 n^3}{1 + \gamma \mu n} \le 1 - \frac{\gamma n \mu}{2}.$$

Let us simplify it:

$$1 + 4\gamma^3 L^3 n^3 \le 1 + \frac{\gamma n \mu}{2} - \frac{\gamma^2 n^2 \mu^2}{2}$$

$$4\gamma^2 L^3 n^3 \le \frac{n \mu}{2} - \frac{\gamma n^2 \mu^2}{2}$$

$$4\gamma^2 L^3 n^2 \le \frac{\mu}{2} - \frac{\gamma n \mu^2}{2}$$

$$4\gamma^2 L^3 n^2 + \frac{\gamma n \mu^2}{2} \le \frac{\mu}{2}.$$

Now as $\gamma \leq \frac{1}{4Ln}\sqrt{\frac{\mu}{L}}$, we have

$$2 \cdot \frac{1}{16L^2n^2} \cdot \frac{\mu}{L}L^3n^2 + \frac{1}{4Ln}\sqrt{\frac{\mu}{L}} \cdot \frac{n\mu^2}{2} \leq \frac{\mu}{2}$$

$$\frac{1}{4}\mu + \frac{1}{8}\frac{\mu}{L}\sqrt{\frac{\mu}{L}}\mu \leq \frac{\mu}{2}$$

$$\frac{1}{4} + \frac{1}{8}\frac{\mu}{L}\sqrt{\frac{\mu}{L}} \leq \frac{1}{2}.$$

It is true since $n \geq 1$ and $\mu \leq L$. This ends proof of Theorem 7.

## F.2 Proof of Theorem 8

Suppose the functions $f_1, f_2, \ldots, f_n$ are convex and Assumption 1 hold.s Then for Algorithm **??** with a stepsize $\gamma \leq \frac{1}{2\sqrt{2}Ln}$, the average iterate $\hat{x}_T := \frac{1}{T}\sum_{j=1}^{T} x_j$ satisfies

$$\mathbb{E}\left[f(\hat{x}_T) - f(x_*)\right] \leq \frac{2\|x_0 - x_*\|^2}{\gamma nT}.$$

We start with Lemma 8 from Mishchenko et al. (2020):

$$\mathbb{E}\left[\|x_{t+1} - x_*\|^2 \mid x_t\right] \leq \|x_t - x_*\|^2 - 2\gamma n\mathbb{E}\left[f(x_{t+1}) - f(x_*) \mid x_t\right] + \gamma^3 Ln^3\sigma_*^2$$

$$2\gamma n\mathbb{E}\left[f(x_{t+1}) - f(x_*) \mid x_t\right] \leq \|x_t - x_*\|^2 - \mathbb{E}\left[\|x_{t+1} - x_*\|^2 \mid x_t\right] + \gamma^3 Ln^3\sigma_*^2.$$

Using Lemma 1 and considering $y_t = x_t$, we have

$$\left(\sigma_*^t\right)^2 \leq 8LD_{f^t}(x_t, x_*).$$

Applying Proposition 1 we get

$$\left(\sigma_*^t\right)^2 \leq 8LD_f(x_t, x_*) = 8L(f(x_t) - f(x_*)).$$

Next, we utilize the inner product reformulation and get

$$2\gamma n\mathbb{E}\left[f(x_{t+1}) - f(x_*) \mid x_t\right] \leq \|x_t - x_*\|^2 - \mathbb{E}\left[\|x_{t+1} - x_*\|^2 \mid x_t\right] + \gamma^3 Ln^3 \cdot 8L(f(x_t) - f(x_*)).$$

Using tower property we have

$$2\gamma n\mathbb{E}\left[f(x_{t+1}) - f(x_*)\right] \leq \mathbb{E}\left[\|x_t - x_*\|^2\right] - \mathbb{E}\left[\|x_{t+1} - x_*\|^2\right] + 8\gamma^3 L^2 n^3\mathbb{E}\left[(f(x_t) - f(x_*))\right].$$

Now we subtract from both sides:

$$2\gamma n\mathbb{E}\left[f(x_{t+1}) - f(x_*)\right] - 8\gamma^3 L^2 n^3\mathbb{E}\left[f(x_{t+1}) - f(x_*)\right] \leq \mathbb{E}\left[\|x_t - x_*\|^2\right] - \mathbb{E}\left[\|x_{t+1} - x_*\|^2\right]$$
$$+ 8\gamma^3 L^2 n^3\mathbb{E}\left[(f(x_t) - f(x_*))\right]$$
$$- 8\gamma^3 L^2 n^3\mathbb{E}\left[f(x_{t+1}) - f(x_*)\right]$$

$$\left(2\gamma n - 8\gamma^3 L^2 n^3\right)\mathbb{E}\left[f(x_{t+1}) - f(x_*)\right] \leq \mathbb{E}\left[\|x_t - x_*\|^2\right] - \mathbb{E}\left[\|x_{t+1} - x_*\|^2\right]$$
$$+ 8\gamma^3 L^2 n^3\left(\mathbb{E}\left[f(x_t) - f(x_*)\right] - \mathbb{E}\left[f(x_{t+1}) - f(x_*)\right]\right)$$

$$2\gamma n\left(1 - 4\gamma^2 L^2 n^2\right)\mathbb{E}\left[f(x_{t+1}) - f(x_*)\right] \leq \mathbb{E}\left[\|x_t - x_*\|^2\right] - \mathbb{E}\left[\|x_{t+1} - x_*\|^2\right]$$
$$+ 8\gamma^3 L^2 n^3\left(\mathbb{E}\left[f(x_t) - f(x_*)\right] - \mathbb{E}\left[f(x_{t+1}) - f(x_*)\right]\right).$$

Summing these inequalities for $t = 0, 1, \ldots, T - 1$ gives

$$2\gamma n \left(1 - 4\gamma^2 L^2 n^2\right) \sum_{t=0}^{T-1} \mathbb{E}\left[f\left(x_{t+1}\right) - f\left(x_*\right)\right] \leq \sum_{t=0}^{T-1} \left(\mathbb{E}\left[\|x_t - x_*\|^2\right] - \mathbb{E}\left[\|x_{t+1} - x_*\|^2\right]\right)$$

$$+ 8\gamma^3 L^2 n^3 \sum_{t=0}^{T-1} \left(\mathbb{E}\left[f(x_t) - f(x_*)\right] - \mathbb{E}\left[f\left(x_{t+1}\right) - f\left(x_*\right)\right]\right)$$

$$= \mathbb{E}\left[\|x_0 - x_*\|^2\right] - \mathbb{E}\left[\|x_T - x_*\|^2\right]$$

$$+ 8\gamma^3 L^2 n^3 \mathbb{E}\left[f\left(x_0\right) - f\left(x_*\right)\right] - 8\gamma^3 L^2 n^3 \mathbb{E}\left[f\left(x_T\right) - f\left(x_*\right)\right]$$

$$\leq \mathbb{E}\left[\|x_0 - x_*\|^2\right] + 8\gamma^3 L^2 n^3 \mathbb{E}\left[f\left(x_0\right) - f\left(x_*\right)\right]$$

$$\leq \mathbb{E}\left[\|x_0 - x_*\|^2\right] + 4\gamma^3 L^3 n^3 \mathbb{E}\left[\|x_0 - x_*\|^2\right]$$

$$= (1 + 4\gamma^3 L^3 n^3)\mathbb{E}\left[\|x_0 - x_*\|^2\right],$$

and dividing both sides by $2\gamma n \left(1 - 4\gamma^2 L^2 n^2\right) T$, we get

$$\frac{1}{T} \sum_{t=0}^{T-1} \mathbb{E}\left[f\left(x_{t+1}\right) - f\left(x_*\right)\right] \leq \frac{1 + 4\gamma^3 L^3 n^3}{1 - 4\gamma^2 L^2 n^2} \frac{\|x_0 - x_*\|^2}{2\gamma n T}.$$

Using the convexity of $f$, the average iterate $\hat{x}_T \stackrel{\text{def}}{=} \frac{1}{T} \sum_{t=1}^{T} x_t$ satisfies

$$\mathbb{E}\left[f\left(\hat{x}_T\right) - f\left(x_*\right)\right] \leq \frac{1}{T} \sum_{t=1}^{T} \mathbb{E}\left[f\left(x_t\right) - f\left(x_*\right)\right] \leq \frac{1 + 4\gamma^3 L^3 n^3}{1 - 4\gamma^2 L^2 n^2} \frac{\|x_0 - x_*\|^2}{2\gamma n T}.$$

Let us show that

$$\frac{1 + 4\gamma^3 L^3 n^3}{1 - 4\gamma^2 L^2 n^2} \leq 4.$$

Applying $\gamma \leq \frac{1}{2\sqrt{2}Ln}$ we have

$$\frac{1 + 4\frac{1}{16\sqrt{2}L^3 n^3}L^3 n^3}{1 - 4\frac{1}{8L^2 n^2}L^2 n^2} = \frac{1 + \frac{1}{4\sqrt{2}}}{1 - \frac{1}{2}} \leq 4.$$

Finally, we have

$$\mathbb{E}\left[f\left(\hat{x}_T\right) - f\left(x_*\right)\right] \leq \frac{2\|x_0 - x_*\|^2}{\gamma n T}.$$

This ends the proof.

## G  One More Algorithm: RR-VR

### G.1  New Algorithm: RR-VR

---

**Algorithm 2** Random Reshuffling with Variance Reduction

---

1: **Input:** Stepsize $\gamma > 0$, probability $p$, $x_0 = x_0^0 \in \mathbb{R}^d$, $y_0 \in \mathbb{R}^d$, number of epochs $T$.
2: **for** $t = 0, 1, \ldots T - 1$ **do**
3:    Choose a random permutation $\{\pi_0, \ldots, \pi_{n-1}\}$ of $\{1, \ldots, n\}$
4:    $x_t^0 = x_t$
5:    **for** $i = 0, \ldots, n - 1$ **do**
6:      $g_t^i(x_t^i, y_t) = \nabla f_{\pi_i}(x_t^i) - \nabla f_{\pi_i}(y_t) + \nabla f(y_t)$
7:      $x_t^{i+1} = x_t^i - \gamma g_t^i(x_t^i, y_t)$
8:    **end for**
9:    $x_{t+1} = x_t^n$
10:   $y_{t+1} = \begin{cases} y_t & \text{with probability } 1 - p \\ x_t & \text{with probability } p \end{cases}$
11: **end for**

---

In this section we formulate convergence results for a generalized version of SVRG under random reshuffling. Analysis of RR-VR (Algorithm 2) is more complicated.

### G.2  Convergence Theory

To analyze this method, we introduce Lyapunov functions.

**Theorem 9.** *Suppose that each $f_i$ is convex, $f$ is $\mu$-strongly convex, and Assumption 1 holds. Then provided the parameters satisfy $n > \kappa$, $\frac{\kappa}{n} < p < 1$ and $\gamma \leq \frac{1}{2\sqrt{2}Ln}$, the final iterate generated by RR-VR (Algorithm 2) satisfies $V_T \leq \max(q_1, q_2)^T V_0$, where $q_1 = 1 - \frac{\gamma\mu n}{4}\left(1 - \frac{p}{2}\right)$, $q_2 = 1 - p + \frac{8}{\mu}\gamma^2 L^3 n$, and the Lyapunov function is defined via*

$$V_t := \mathbb{E}\left[\|x_t - x_*\|^2\right] + \left(\frac{4}{\gamma\mu n}\right)^{-1} \mathbb{E}\left[\|y_t - x_*\|^2\right].$$

*This means that the iteration complexity of Algorithm 2 is $T = O\left(\kappa \log\left(\frac{1}{\varepsilon}\right)\right)$.*

Note that the probability $p$ should not be too small. We obtain the same complexity as that of of Rand-Reshuffle.

**Theorem 10.** *Suppose that the functions $f_1, \ldots, f_n$ are $\mu$-strongly convex, and that Assumption 1 holds. Then for RR-VR (Algorithm 2) with parameters that satisfy $\gamma \leq \frac{1}{2L}\sqrt{\frac{\mu}{2nL}}$, $\frac{1}{2} < \delta < \frac{1}{\sqrt{2}}$, $0 < p < 1$, and for a sufficiently large number of functions, $n > \log\left(\frac{1}{1-\delta^2}\right) \cdot \left(\log\left(\frac{1}{1-\gamma\mu}\right)\right)^{-1}$, the iterates generated by the RR-VR algorithm satisfy $V_T \leq \max(q_1, q_2)^T V_0$, where $q_1 = (1 - \gamma\mu)^n + \delta^2$, $q_2 = 1 - p\left(1 - \frac{2\gamma^2 L^3 n}{\mu\delta^2}\right)$, and*

$$V_t := \mathbb{E}\left[\|x_t - x_*\|^2\right] + \frac{\delta^2}{p}\mathbb{E}\left[\|y_t - x_*\|^2\right].$$

*This means that the iteration complexity of Algorithm 2 is $T = O\left(\max\left(\kappa\sqrt{\frac{\kappa}{n}}, \frac{1}{2\log(2\delta)}\right)\log\left(\frac{1}{\varepsilon}\right)\right)$.*

We get almost the same rate as the rate of Rand-Reshuffle, but there is one difference. Complexity depends on $\delta$ term. However, the first term dominates in most cases.

### G.3  Proof of Theorem 9

Suppose that each $f_i$ is convex, $f$ is $\mu$-strongly convex, and Assumption 1 holds. Then provided the parameters satisfy $n > \kappa$, $\frac{\kappa}{n} < p < 1$ and $\gamma \leq \frac{1}{2\sqrt{2}Ln}$, the final iterate generated by RR-VR (Algorithm 2) satisfies

$$V_T \leq \max(q_1, q_2)^T V_0,$$

where

$$q_1 = 1 - \frac{\gamma\mu n}{4}\left(1 - \frac{p}{2}\right), \quad q_2 = 1 - p + \frac{8}{\mu}\gamma^2 L^3 n,$$

and the Lyapunov function is defined via

$$V_t := \mathbb{E}\left[\|x_t - x_*\|^2\right] + \frac{4}{\gamma\mu n}\mathbb{E}\left[\|y_t - x_*\|^2\right].$$

*Proof.* For the problem $\frac{1}{n}\sum_{i=1}^{n} f_i^t(x)$ we will use an inequality from Mishchenko et al. (2020):

$$\mathbb{E}\left[\|x_{t+1} - x_*\|^2 \mid x_t\right] \leq \frac{1}{1 + \gamma\mu n}\left(\|x_t - x_*\|^2 + \frac{\gamma^3 L n^2 \sigma_*^2}{2}\right)$$

$$= \frac{1}{1 + \gamma\mu n}\|x_t - x_*\|^2 + \frac{1}{1 + \gamma\mu n}\frac{\gamma^3 L n^2 \sigma_*^2}{2}$$

$$\leq \left(1 - \frac{\gamma\mu n}{2}\right)\|x_t - x_*\|^2 + \frac{\gamma^3 L n^2 \sigma_*^2}{2}.$$

Now we apply inequality

$$\mathbb{E}\left[\|x_{t+1} - x_*\|^2 \mid x_t, y_t\right] \leq \left(1 - \frac{\gamma\mu n}{2}\right)\|x_t - x_*\|^2 + \frac{\gamma^3 L n^2 \sigma_*^2}{2}$$

$$\leq \left(1 - \frac{\gamma\mu n}{2}\right)\|x_t - x_*\|^2 + 2\gamma^3 L^3 n^2 \|y_t - x_*\|^2.$$

Using tower property we have

$$\mathbb{E}\left[\|x_{t+1} - x_*\|^2\right] = \mathbb{E}\left[\mathbb{E}\left[\|x_{t+1} - x_*\|^2 \mid x_t, y_t\right]\right]$$

$$\leq \left(1 - \frac{\gamma\mu n}{2}\right)\mathbb{E}\left[\|x_t - x_*\|^2\right] + 2\gamma^3 L^3 n^2 \mathbb{E}\left[\|y_t - x_*\|^2\right].$$

Now we look at

$$y_{t+1} = \begin{cases} y_t & \text{with probability } 1 - p \\ x_t & \text{with probability } p \end{cases}.$$

We get

$$\mathbb{E}\left[\|y_{t+1} - x_*\|^2 \mid x_t, y_t\right] = (1 - p)\|y_t - x_*\|^2 + p\|x_t - x_*\|^2.$$

Using tower property

$$\mathbb{E}\left[\|y_{t+1} - x_*\|^2\right] = \mathbb{E}\left[\mathbb{E}\left[\|y_{t+1} - x_*\|^2 \mid x_t, y_t\right]\right]$$

$$= (1 - p)\mathbb{E}\left[\|y_t - x_*\|^2\right] + p\mathbb{E}\left[\|x_t - x_*\|^2\right].$$

Finally, we have

$$\mathbb{E}\left[\|x_{t+1} - x_*\|^2\right] + M\mathbb{E}\left[\|y_{t+1} - x_*\|^2\right] \leq \left(1 - \frac{\gamma\mu n}{2}\right)\|x_t - x_*\|^2 + 2\gamma^3 L^3 n^2 \mathbb{E}\left[\|y_t - x_*\|^2\right]$$

$$+ (1 - p)M\mathbb{E}\|y_t - x_*\|^2 + pM\mathbb{E}\|x_t - x_*\|^2.$$

Denote $V_t = \mathbb{E}\left[\|x_t - x_*\|^2\right] + M\mathbb{E}\left[\|y_t - x_*\|^2\right]$. Using this we obtain

$$V_{t+1} \leq \left(1 - \frac{\gamma\mu n}{2}\right)\mathbb{E}\left[\|x_t - x_*\|^2\right] + 2\gamma^3 L^3 n^2 \mathbb{E}\left[\|y_t - x_*\|^2\right]$$

$$+ (1 - p)M\mathbb{E}\left[\|y_t - x_*\|^2\right] + pM\mathbb{E}\left[\|x_t - x_*\|^2\right].$$

Thus,

$$V_{t+1} \leq \left(1 - \frac{\gamma\mu n}{2} + pM\right)\mathbb{E}\left[\|x_t - x_*\|^2\right] + \left(1 - p + \frac{1}{M}2\gamma^3 L^3 n^2\right) M\mathbb{E}\left[\|y_t - x_*\|^2\right].$$

To have contraction we use

$$M = \frac{\gamma\mu n}{4}, \qquad \gamma = \frac{1}{2\sqrt{2}Ln}.$$

We have the final rate

$$V_{t+1} \leq \max\left(1 - \frac{\gamma\mu n}{4}\left(1 - \frac{p}{2}\right), 1 - p + \frac{8}{\mu}\gamma^2 L^3 n\right) V_t$$

$$V_T \leq \max\left(1 - \frac{\gamma\mu n}{4}\left(1 - \frac{p}{2}\right), 1 - p + \frac{8}{\mu}\gamma^2 L^3 n\right)^T V_0.$$

$\square$

## G.4   Proof of Theorem 10

Suppose that the functions $f_1, \ldots, f_n$ are $\mu$-strongly convex, and that Assumption 1 holds. Then for RR-VR (Algorithm 2) with parameters that satisfy $\gamma \leq \frac{1}{2L}\sqrt{\frac{\mu}{2nL}}$, $\frac{1}{2} < \delta < \frac{1}{\sqrt{2}}$, $0 < p < 1$, and for a sufficiently large number of functions, $n > \log\left(\frac{1}{1-\delta^2}\right) \cdot \left(\log\left(\frac{1}{1-\gamma\mu}\right)\right)^{-1}$, the iterates generated by the RR-VR algorithm satisfy

$$V_T \leq \max\left(q_1, q_2\right)^T V_0,$$

where

$$q_1 = (1 - \gamma\mu)^n + \delta^2, \quad q_2 = 1 - p\left(1 - \frac{2\gamma^2 L^3 n}{\mu\delta^2}\right),$$

and

$$V_t := \mathbb{E}\left[\|x_t - x_*\|^2\right] + \frac{\delta^2}{p}\mathbb{E}\left[\|y_t - x_*\|^2\right].$$

*Proof.* For the problem $\frac{1}{n}\sum_{i=1}^{n} f_i^t(x)$ we will use two inequalities from Mishchenko et al. (2020):

$$\mathbb{E}\left[\|x_{t+1} - x_*\|^2 \mid x_t\right] \leq (1 - \gamma\mu)^n \|x_t - x_*\|^2 + 2\gamma^2\sigma_{\text{Shuffle}}^2\left(\sum_{i=0}^{n-1}(1 - \gamma\mu)^i\right)$$

$$\sigma_{\text{Shuffle}}^2 \leq \frac{\gamma Ln}{4}\sigma_*^2.$$

Using this result, we have

$$\mathbb{E}\left[\|x_{t+1} - x_*\|^2 \mid x_t, y_t\right] \leq (1 - \gamma\mu)^n \|x_t - x_*\|^2 + \frac{1}{2}\gamma^3 Ln\sigma_*^2\left(\sum_{i=0}^{n-1}(1 - \gamma\mu)^i\right)$$

$$\leq (1 - \gamma\mu)^n \|x_t - x_*\|^2 + \frac{1}{\mu}2\gamma^2 L^2 nL\|y_t - x_*\|^2.$$

Using tower property

$$\mathbb{E}\left[\|x_{t+1} - x_*\|^2\right] = \mathbb{E}\left[\mathbb{E}\left[\|x_{t+1} - x_*\|^2 \mid x_t, y_t\right]\right]$$

$$\leq (1 - \gamma\mu)^n \mathbb{E}\left[\|x_t - x_*\|^2\right] + \frac{1}{\mu}2\gamma^2 LnL^2\mathbb{E}\left[\|y_t - x_*\|^2\right].$$

Now we look at

$$y_{t+1} = \begin{cases} y_t & \text{with probability } 1 - p \\ x_t & \text{with probability } p \end{cases}.$$

Thus, $\mathbb{E}\left[\|y_{t+1} - x_*\|^2 \mid x_t, y_t\right] = (1 - p)\|y_t - x_*\|^2 + p\|x_t - x_*\|^2$. Using tower property

$$\mathbb{E}\left[\|y_{t+1} - x_*\|^2\right] = \mathbb{E}\left[\mathbb{E}\left[\|y_{t+1} - x_*\|^2 \mid x_t, y_t\right]\right]$$
$$= (1 - p)\mathbb{E}\left[\|y_t - x_*\|^2\right] + p\mathbb{E}\left[\|x_t - x_*\|^2\right].$$

Denote $V_t = \mathbb{E}\left[\|x_t - x_*\|^2\right] + M\mathbb{E}\left[\|y_t - x_*\|^2\right]$ and we have

$$V_{t+1} = \mathbb{E}\left[\|x_{t+1} - x_*\|^2\right] + M\mathbb{E}\left[\|y_{t+1} - x_*\|^2\right]$$
$$\leq (1 - \gamma\mu)^n \mathbb{E}\left[\|x_t - x_*\|^2\right] + \frac{2}{\mu}\gamma^2 L^3 n\mathbb{E}\left[\|y_t - x_*\|^2\right] + (1 - p)M\mathbb{E}\left[\|y_t - x_*\|^2\right] + pM\mathbb{E}\left[\|x_t - x_*\|^2\right]$$
$$\leq ((1 - \gamma\mu)^n + pM)\mathbb{E}\left[\|x_t - x_*\|^2\right] + \left((1 - p) + \frac{2\gamma^2 L^3 n}{\mu M}\right)M\mathbb{E}\left[\|x_t - x_*\|^2\right]$$
$$\leq \max\left(((1 - \gamma\mu)^n + pM), \left((1 - p) + \frac{2\gamma^2 L^3 n}{\mu M}\right)\right)V_t.$$

Unrolling the recusrion we have

$$V_T \leq \max\left(((1 - \gamma\mu)^n + pM), \left(1 - p + \frac{2\gamma^2 L^3 n}{\mu M}\right)\right)^T V_0.$$

Applying $M = \frac{\delta^2}{p}$ and $\gamma \leq \frac{1}{2L}\sqrt{\frac{\mu}{2nL}}$ we get

$$V_T \leq \max\left((1 - \gamma\mu)^n + \delta^2, 1 - p\left(1 - \frac{2\gamma^2 L^3 n}{\mu\delta^2}\right)\right)^T V_0.$$

$\square$

