# OpenReview forum: "Better Linear Rates for SGD with Data Shuffling"
_TMLR — Rejected by TMLR_

### Review · Reviewer_JuYT · 2022-09-19

**Summary Of Contributions:**

This paper provides a convergence rate analysis on variance-reduced stochastic optimization methods, which adopt sampling without replacement strategies, including random shuffling and random reshuffling, for finite-sum minimization problems.
Utilizing techniques developed in [Mishchenko et al. (2020)], the authors showed the linear convergence rates for strongly convex settings and provided several convergence analyses for other settings.
The obtained linear convergence rates are faster than those of other methods based on sampling without replacement.

**Requested Changes:**

The technical issues for random shuffling strategy should be fixed. If my understanding is wrong, I would appreciate it if the authors could correct me.

**Strengths And Weaknesses:**

**Strengths**

Almost theoretical studies on stochastic optimization methods consider a sampling with replacement strategy, whereas we empirically use a sampling without replacement strategy. Therefore, convergence analysis for random (re-)shuffling strategy has recently become an important research topic. In this sense, this study contributes by improving known convergence rates for strongly convex problems.
The proof technique is very interesting. The key is to consider surrogate loss functions which are consistent with the update rule of the methods, which cause the variance reduction.

**Weaknesses**

I basically like the idea behind this analysis, but I think there is a technical flaw in the proof for the random-shuffling strategy in which the random shuffle is conducted only at the beginning of the training.
Thus, at least a major revision is needed.

- Theorem 1 and 2 build upon Lemma 3 in [Mishchenko et al. (2020)]. However, Lemma 3 probably does not hold for the random shuffling strategy for the proposed method because of the use of surrogate functions. The evaluation of $\mathbb{E}[ || \sum_{i=k}^{n-1} \nabla f_{\pi_i}(x_*)||^2 ]$ is the key in Lemma 3 ([Mishchenko et al. (2020)]), but this evaluation cannot apply to $\mathbb{E}[ || \sum_{i=k}^{n-1} \nabla f_{\pi_i}^t (x_*)||^2 ]$ when using random shuffling because the function $f_i^t$ itself depends on the history of optimization (i.e., $f_i^t$ depends on $\pi_i$ as well). This does not matter for a random reshuffling strategy because the same evaluation in Lemma 3 can apply to the conditional expectation.
- A similar issue happens in the proof of Theorem 3 in which an inequality $(\sigma_{\mathrm{shuffle}}^t)^2 \leq \frac{\gamma L n}{4} (\sigma_*^t)^2$ is used. (Note the superscript $t$ is missing.) But, this inequality probably does not hold for random shuffling for the same reason as above.

The above issues happen in several places.
Moreover, some typos confuse the readers.
- Eq. (4): If this definition of $a_t^i$ and Eq. (3) are correct, then $\nabla f_{\pi_i}^t(x)$ (in Eq. (5)) is $\nabla f_{\pi_i}(x) - \nabla f_{\pi_{\pi_i}}(y_t) + \nabla f(y_t)$, which contradict Eq. (5).  I guess $a_t^i = -\nabla f_i(y_t) + \nabla f(y_t)$ is correct (?) The same typo appears on page 24.
- The superscript $t$ is missing in the proof of Theorem 3.

---

### Review · Reviewer_UE4F · 2022-09-20

**Summary Of Contributions:**

This paper provides the analysis of SVRG when applied to SGD with Data Shuffling in
1. Four settings: Non-convex, convex, and strongly convex objectives (with and without strongly convex component functions); and
2. For three variants: deterministic shuffling, random shuffling and random reshuffling.

The paper is the first to prove convergence bounds in some settings, and matching or better bounds than existing in others.

Even though the theoretical bounds are better than existing ones, the paper says that in some settings they are still not better than the convergence bounds for vanilla SVRG (where sampling is done with-replacement). This shows that there is still a gap between theory and practice, where we see that Data Shuffling based SGD (that is, without-replacement SGD) is better than with-replacement SGD.

**Broader Impact Concerns:**

I do not see any concerns regarding the ethical implications of this paper.

**Requested Changes:**

1. I feel that for readers familiar with SVRG, equation (5) is easily understood as a direct analog. Hence in Section 2.1, it would be easier (for the readers) if the formulation of equation (5) is proposed directly as an analog of SVRG for data shuffling SGD method. However, I leave this up to the authors if they want to make this change.

2. The paper heavily cites Mishchenko et al. (2020) and credits it (solely) for providing tight convergence rates for shuffling based SGD. However, the concurrent work of Ahn et al. (2020) is missing, which seems to be doing the same thing. Please add citations for the paper, and if the authors believe that Ahn et al. (2020)'s analysis had limitations as compared to Mishchenko et al. (2020), please specify those too.

3. While comparing to the results of Huang et al. (2021) in table 1, please mention that $L^2/\epsilon$ is the complexity when the error $\epsilon$ is measured in gradient difference norm. This is inconsistent because this paper measures the error metric as the function value difference. Hence, the results cannot be directly compared.

4. Please also add comparison with Beznosikov (2021)

Ahn, Kwangjun, Chulhee Yun, and Suvrit Sra. SGD with shuffling: optimal rates without component convexity and large epoch requirements. 2020
Beznosikov, Aleksandr, and Martin Takáč. Random-reshuffled SARAH does not need a full gradient computations. 2021

**Strengths And Weaknesses:**

Strengths:
1. As mentioned in the summary section, the paper covers all the major variants and function settings in its analysis.
2. Further, the paper is the first to prove convergence bounds in some settings, and matching or better bounds than existing in others (except the case when components are strongly convex and the number of samples is less than the condition number. Then Prox-DFinito seems to be better). This is especially significant considering that it has been difficult to prove convergence bounds for shuffling based SGD.
3. Another advantage of the paper is that since it proves bounds for SVRG on SGD with shuffling, the memory requirement is lesser as compared to the ones that use versions of the Finito algorithm.
4. The paper is well written.

Weaknesses:
The paper discusses in Section 2.5 that while in practice we see without-replacement sampling based SGD to be faster than with-replacement one, the bounds proved in this paper do not reflect that. Hence, the bounds proved in this paper are probably not tight. However, Safran & Shamir (2021) have provided evidence that in general, sampling without-replacement might not be better than its with-replacement counterpart. I suspect to truly prove better convergence for without-replacement sampling, one might need distributional assumptions on samples, which reflect the distribution of the data we see in practice.

Safran, Itay, and Ohad Shamir. Random Shuffling Beats SGD Only After Many Epochs on Ill-Conditioned Problems. 2021

---

### Review · Reviewer_36ye · 2022-09-23

**Summary Of Contributions:**

In most of the analysis, sampling with replacement is used for  standard and variance-reduced variants of SGD. Most ML methods use sampling without replacement, which has been found empirically better and therefore acts as the de facto default sampling mechanism in Deep Learning. The training data is sampled exactly once by epoch, where an epoch is obtained using a random permutation of the training data.
(i) In the deterministic shuffling (det-shuffle), the training data are processed cyclically in a natural order. That is, a deterministic permutation is used throughout the training process.
(ii) In random shuffling, the training data is instead randomly shuffled/distributed. This is done only once, before the training process begins, and the selection of training data then follows a cyclic pattern.  There are almost no non-trivial analyzes for this method (Mishchenko et al., 2020). This strategy works very well in practice.
(iii) In random reshuffling (Rand-Reshuffle), the training data are randomly reshuffled before the start of each epoch (edge reshuffle). Its empirical performance is often very similar to that of Rand-Shuffle.

The main idea of the paper to perturb the objective function with $n$ nonzero linear functions (adding to zero). The perturbation is applied at the beginning of each epoch, Denote $a_t^i, \ldots, a_t^n \in \mathbb{R}^d$ summing up to zero: $\sum_{i=1}^n a_t^i=0$. Let $a^t=\left(a_t^i, \ldots, a_t^n\right)$. Adding this structured zero to $f$, the problem is reformulated
$$
f(x)=\frac{1}{n} \sum_{i=1}^n f_i(x)=\frac{1}{n} \sum_{i=1}^n\left(f_i(x)+\left\langle a_t^i, x\right\rangle\right):=\frac{1}{n} \sum_{i=1}^n f_i^t(x)
$$
where $f_i^t(x):=f_i(x)+\left\langle a_t^i, x\right\rangle$. Note that
$$
\nabla f_i^t(x)=\nabla f_i(x)+a_t^i
$$
The key proposal of this paper  is to run  Det-Shuffle, Rand-Shuffle and Rand-Reshuffle methods,  but in each epoch to apply them to the current reformulated problem
$
\min _{x \in R^d} \frac{1}{n} \sum_{i=1}^n f_i^t(x) .
$

**Broader Impact Concerns:**

on this side, it is perfectly clear, there is nothing to fear!

**Requested Changes:**

- There are some mysterious sentences : We also believe that our theoretical results can be applied to other aspects of machine learning, leading to improvements in state of the art for current or future applications": which ones ?
- I am not sure to understand: " However, when seen that way, we do not observe an improvement in complexity. The reason for this is that there is a large gap in our understanding of shuffling based methods, especially for variance reduced variants, which does not yet allow for theoretical speedups compared to their sampling-with-replacement cousins." Is this a problem related to a "sub-optimal" analysis, or is there no difference in practice between sampling methods with and without replacement when using a variance reduction method?
- The logarithmic scale is a bit misleading in figure 2... We start to see differences between $10^{-10}$ and $10^{-12}$ so we don't care much, do we?
- I would have liked some comments under figures 3 and 4. A little interpretation would help here
- Missing reference in the proof of Theorem 8 ?


**Strengths And Weaknesses:**

Strengths
- The paper is very well written, a pleasure to read, and it makes it very easy to follow the derivations that are not earth-shatteringly original, but nonetheless non-trivial.
- The results are clearly stated and improve on the state of the art
- The results are stated for smooth, (strongly) convex and non-convex functions.
- compare the variance reduced algorithms with and without reshuffling
Weaknesses:
- The message is  unclear. What I have understood is that Rand-Shuffle and Rand Reshuffle outperformed RR-SAGA anf AVRG. Variance reduction coupled with Randon Shuffling or Random Reshuffling degrades the performance: this should be more clearly stressed.
- The paper thus improves on the Mishchencko et al (2020) paper [if I understand correctly], retrieves the results already established in Ying et al (2020) for variance reduction [which are worse than the versions without variance reduction]. The essential originality is the analysis of Det -Shuffle in the convex and strongly convex case... It's not very thick though!
- I am not a fan of SAGA.... who is still interested in an algorithm whose storage cost is $dn$ ?  SAGA is the archetype of the completely fanciful algorithm that is only of theoretical interest.
- Experiments could be more interesting and conclusive

---

### Comment · Action_Editors · 2022-10-09
**Discussion**

Dear authors,

The three reviews are available since September 23. If you want to engage in a discussion with the reviewers and update the manuscript, please start doing so as soon as possible as reviewers are expected to submit their decision recommendation within 1-2 weeks.

Thank you

---

### Decision · Action_Editors · 2022-11-04

**Recommendation:** Reject

**Comment:**

The authors did not provide any clarifications or corrections regarding the potential problem in their proof pointed out by one reviewer. Because of this and despite an otherwise positive general sentiment towards the paper, all reviewers recommend to reject the paper. I agree that the paper should be rejected in its current form.

Given the above circumstances, I would be willing to consider a significantly revised version of the manuscript for resubmission to TMLR, if such a revised version appropriately clarifies and/or corrects the involved technical derivations.

**Audience:**

Yes, this paper improves existing convergence analysis of SGD with sampling without replacement, in particular the ones based on data shuffling strategies. These are most commonly used by practitioners, and it is thus interesting for the machine learning community to better understand the convergence rates of such methods. While the work is somewhat incremental with respect to previously published works, it does improve upon them in some regimes.

**Claims And Evidence:**

One reviewer pointed out a possible problem in the proofs, which, if confirmed, would affect the results and thus the evidence and claims made in the paper.